# On the Efficiency of ERM in Feature Learning

**Ayoub El Hanchi**
University of Toronto &
Vector Institute
aelhan@cs.toronto.edu

**Chris J. Maddison**
University of Toronto &
Vector Institute
cmaddis@cs.toronto.edu

**Murat A. Erdogdu**
University of Toronto &
Vector Institute
erdogdu@cs.toronto.edu

## Abstract

Given a collection of feature maps indexed by a set $\mathcal{T}$, we study the performance of empirical risk minimization (ERM) on regression problems with square loss over the union of the linear classes induced by these feature maps. This setup aims at capturing the simplest instance of feature learning, where the model is expected to jointly learn from the data an appropriate feature map and a linear predictor. We start by studying the asymptotic quantiles of the excess risk of sequences of empirical risk minimizers. Remarkably, we show that when the set $\mathcal{T}$ is not too large and when there is a unique optimal feature map, these quantiles coincide, up to a factor of two, with those of the excess risk of the oracle procedure, which knows a priori this optimal feature map and deterministically outputs an empirical risk minimizer from the associated optimal linear class. We complement this asymptotic result with a non-asymptotic analysis that quantifies the decaying effect of the global complexity of the set $\mathcal{T}$ on the excess risk of ERM, and relates it to the size of the sublevel sets of the suboptimality of the feature maps. As an application of our results, we obtain new guarantees on the performance of the best subset selection procedure in sparse linear regression under general assumptions.

## 1 Introduction

A central idea in modern machine learning is that of data-driven feature learning. Specifically, instead of performing linear prediction on top of handcrafted features, the current dominant paradigm suggests to use models that select useful features for linear prediction in a data-dependent way [e.g. KSH12; LBH15; He+16; Vas+17]. Of course, by putting the burden of picking a feature map on the model and data, we should expect that the resulting learning problem will require more samples to be solved. But just how many more samples do we need to learn such feature-learning-based models?

In this paper, we investigate this question in a general setting. We study the performance of empirical risk minimization (ERM) on regression tasks with square loss and over model classes induced by arbitrary collections of features maps. More precisely, let $X$ be the random input taking value in a set $\mathcal{X}$, and let $(\phi_t)_{t \in \mathcal{T}}$, $\phi_t : \mathcal{X} \to \mathbb{R}^d$, be a collection of feature maps indexed by a set $\mathcal{T}$. For a given regression task and i.i.d. samples, our aim is to understand the performance of ERM over the class of predictors $\cup_{t \in \mathcal{T}} \{ x \mapsto \langle w, \phi_t(x) \rangle \mid w \in \mathbb{R}^d \}$ as a function of the sample size, the distribution of the data, and relevant properties of the collection of feature maps $(\phi_t)_{t \in \mathcal{T}}$.

Classical uniform-convergence-based analyses would suggest that the performance of ERM in this setting is determined by the size of the model class, appropriately measured. The main message of this paper is that in this case, this is wrong in a strong sense. Specifically, we prove an upper bound on the excess risk of ERM on this problem whose dependence on the size of the model class decays monotonically with the sample size, and eventually depends only on the size of the model class induced by the collection of optimal feature maps, which is typically much smaller.

38th Conference on Neural Information Processing Systems (NeurIPS 2024).

**Formal setup.** We briefly formalize our problem here. Let $X$ be the random input taking value in a set $\mathcal{X}$, and let $(\phi_t)_{t \in \mathcal{T}}$, $\phi_t : \mathcal{X} \to \mathbb{R}^d$, be a collection of feature maps indexed by a set $\mathcal{T}$.[1] Let $Y \in \mathbb{R}$ be the output random variable, jointly distributed with the input $X$. Our goal is to learn to predict the output $Y$ given the input $X$ as well as possible within the class of predictors $\left\{ x \mapsto \langle w, \phi_t(x) \rangle \mid (t, w) \in \mathcal{T} \times \mathbb{R}^d \right\}$. We evaluate the quality of a single prediction $\hat{y}$ given the ground truth $y$ through the loss function $\ell(\hat{y}, y) := (\hat{y} - y)^2 / 2$, and the overall quality of a predictor $(t, w) \in \mathcal{T} \times \mathbb{R}^d$ through its risk

$$R(t, w) := \mathrm{E}[\ell(\langle w, \phi_t(X) \rangle, Y)], \qquad R_* := \inf_{(t,w) \in \mathcal{T} \times \mathbb{R}^d} R(t, w).$$

We assume that we have access to $n$ i.i.d. samples $(X_i, Y_i)_{i=1}^n$ with the same distribution as $(X, Y)$, and perform empirical risk minimization

$$(\hat{t}_n, \hat{w}_n) \in \operatorname*{argmin}_{(t,w) \in \mathcal{T} \times \mathbb{R}^d} R_n(t, w) \quad \text{where} \quad R_n(t, w) := n^{-1} \sum_{i=1}^n \ell(\langle w, \phi_t(X_i) \rangle, Y_i).$$

Our goal is to characterize the excess risk $\mathcal{E}(\hat{t}_n, \hat{w}_n) := R(\hat{t}_n, \hat{w}_n) - R_*$.

**Related work.** The study of upper bounds on the excess risk of ERM in a general setting is a classical topic. It was initiated by Vapnik and Chervonenkis [VC74] who established a link between the excess risk of ERM and the uniform convergence of the underlying empirical process. More recently, and fuelled by the development of Talagrand's concentration inequality [Tal96] and its refinements [e.g. BLM00; Bou02], a literature emerged that provided more fine-grained control of the excess risk of ERM [e.g. BBM05; Kol06; BM06]. A key idea emerging from this line of work is localization. This concept, and in particular the iterative localization method of Koltchinskii [Kol06], plays an important role in our development. We refer the reader to the books [Kol11; Wai19], as well as the recent articles [LRS15; KRV22] for more on this idea.

Focusing on the task of regression with square loss, upper bounds on the excess risk of ERM are available for many classes of predictors, including finite [e.g. Aud07; JRT08; LM09], linear [e.g. LM16b; Oli16; Mou22], and convex classes [e.g. LM16a; Men14; LRS15]. A key development in this area over the last decade has been the realization that such bounds can be obtained under much weaker assumptions than previously thought, owing to the fact that only one-sided control of a certain empirical process is needed, and which can be obtained under very weak assumptions [Men14; KM15; Oli16]. The line of work most closely related to ours is the one on random-design linear regression [AC11; HKZ12; Oli16; LM16b; Sau18; Mou22; EE23], and we view our work as an extension of this literature. We review these results in more detail in Section 2.

Finally, and on a more conceptual level, our work is related to the recent effort to understand the effect of feature learning on the performance of neural networks [e.g. Bac17; Gho+20; Ba+22]. Beyond this conceptual connection however, our work is quite distinct from this literature. Among other things, our setting is more general since we consider arbitrary features maps. In the same vein, it is worth mentioning the line of work on multiple kernel learning [e.g. Lan+04; GA11; SD16], although we are not aware of results from this literature that are directly relevant to our setup.

**Challenges.** Our class of predictors is somewhat unstructured (e.g. it is in general non-convex), so that off-the-shelf results from the above literature are not directly applicable. Nevertheless, the analysis of the performance of ERM on linear classes provides a good starting point as we review in Section 2. Compared to that setting however, we are faced with two additional challenges. First, we need to control an additional source of error arising from the fact that ERM might select a suboptimal feature map. Second, we are lead to study the suprema of certain $\mathcal{T}$-indexed empirical processes, which in the linear setting reduce to single random variables that are easily dealt with.

**Organization.** The rest of the paper is organized as follows. In Section 2, we review known results on the excess risk of linear regression under square loss. In Section 3, we state our main results that hold for the excess risk of ERM for general index sets $\mathcal{T}$. In Section 4, we specialize our analysis to the case where the index set $\mathcal{T}$ is finite, obtain more explicit guarantees, and discuss their implications on the sparse linear regression problem. We conclude in Section 5 with a brief discussion.

---

[1] We assume without loss of generality that if $t, s \in \mathcal{T}$ with $t \neq s$, then $\phi_t$ and $\phi_s$ induce different linear classes of functions, i.e. there is no matrix $A$ such that $\phi_t(x) = A\phi_s(x)$ for all $x \in \mathcal{X}$.

## 2 Background

The goal of this section is to provide more context for our results. We review known results on the excess risk of ERM over linear classes, which corresponds in our setting to the special case where the set $\mathcal{T}$ indexing the feature maps is a singleton. As such, to avoid introducing further notation, we use the one from the previous section, while dropping the dependence on $t$ whenever it occurs.

In the setting of linear regression with square loss, and when the sample covariance matrix of the feature map is invertible, there is a unique empirical risk minimizer and its excess risk admits an explicit expression. Specifically, define

$$\Sigma := \mathrm{E}\big[\phi(X)\phi(X)^T\big], \qquad \Sigma_n := \frac{1}{n}\sum_{i=1}^{n}\phi(X_i)\phi(X_i)^T,$$

and let $w_*$ denote the unique minimizer of the risk $R(w)$.[2] Then, an elementary calculation shows that when $\Sigma_n$ is invertible, there is a unique empirical risk minimizer and it satisfies

$$\hat{w}_n = w_* - \Sigma_n^{-1}\nabla R_n(w_*). \tag{1}$$

Furthermore, since the risk is a quadratic function of $w$ whose gradient at $w_*$ vanishes, replacing $R(\hat{w}_n)$ by the equivalent exact second order Taylor expansion around $w_*$ yields

$$R(\hat{w}_n) - R(w_*) = \frac{1}{2}\|\hat{w}_n - w_*\|_\Sigma^2 = \frac{1}{2}\big\|\Sigma_n^{-1}\nabla R_n(w_*)\big\|_\Sigma^2. \tag{2}$$

While exact, this expression is not readily interpretable. For example, how fast does this excess risk go to 0 as a function of the sample size? The following classical result from asymptotic statistics [e.g. Whi82; LC06; Vaa98] makes this rate more explicit. To state it, we define

$$g(X,Y) := \nabla_w \ell(\langle w_*, \phi(X)\rangle, Y), \qquad G := \mathrm{E}\big[g(X,Y)g(X,Y)^T\big].$$

**Theorem 1.** *Assume that for all $j \in [d]$, $\mathrm{E}\big[\phi_j^2(X)\big] < \infty$, $\mathrm{E}\big[Y^2\big] < \infty$, and $\mathrm{E}[\|g(X,Y)\|_{\Sigma^{-1}}^2] < \infty$. Then, as $n \to \infty$,*

$$n \cdot \mathcal{E}(\hat{w}_n) \xrightarrow{d} \frac{1}{2} \cdot \|Z\|_2^2,$$

*where $Z \sim \mathcal{N}(0, \Sigma^{-1/2}G\Sigma^{-1/2})$. In particular, for any $\delta \in (0, 0.1)$,*

$$\lim_{n\to\infty} n \cdot Q_{\mathcal{E}(\hat{w}_n)}(1-\delta) \asymp \mathrm{E}[\|g(X,Y)\|_{\Sigma^{-1}}^2] + 2\lambda_{\max}(\Sigma^{-1/2}G\Sigma^{-1/2})\log(1/\delta),$$

*where $Q_X(p) := \inf\{x \in \mathbb{R} \mid \mathrm{P}(X \leq x) \geq p\}$ is the quantile function of a random variable $X$, and where we write $a \asymp b$ to mean that there exists absolute constants $C, c$ such that $c \cdot b \leq a \leq C \cdot b$. In the above statement, they can be taken as $C = 1$ and $c = 1/32$.*

We provide a proof in Appendix A for completeness. For our purposes, this theorem is most easily interpreted as follows: for large enough $n$ and small enough $\delta$, if the excess risk of ERM is bounded by some quantity with probability at least $1 - \delta$, then this quantity is, up to a constant, at least as large as the right-hand side of the second displayed equation divided by $n$. While our primary interest is in non-asymptotic bounds, this asymptotic result, by virtue of its exactness, provides us with a benchmark against which such bounds can be compared. In particular, it identifies the quantity $\mathrm{E}[\|g(X,Y)\|_{\Sigma^{-1}}^2]$ as an intrinsic parameter determining the excess risk of ERM on this problem.

For large enough $n$, Theorem 1 gives an interpretable expression for the excess risk. However, it says nothing about how large $n$ needs to be for this expression to be accurate. This motivates a non-asymptotic analysis of the excess risk of ERM, which has been carried out numerous times in recent years [e.g. Oli16; LM16b; EE23]. A goal of this literature has been to obtain upper bounds on the excess risk of ERM that hold in probability under weak moment assumptions, building on the observation that this is indeed possible [Men14]. The following theorem is comparable to the best known result in this area. We leave the proof to Appendix B. To state it, we define

$$V := \mathrm{E}\bigg[\Big(\Sigma^{-1/2}\phi(X)\phi(X)^T\Sigma^{-1/2} - I\Big)^2\bigg], \quad L := \sup_{v \in S^{d-1}} \mathrm{E}\bigg[\Big(\langle v, \Sigma^{-1/2}\phi(X)\rangle^2 - 1\Big)^2\bigg].$$

---

[2]Throughout, we assume without loss of generality that the support of the distribution of $\phi(X)$ is not contained in any hyperplane, which implies the invertibility of $\Sigma$ and the uniqueness of $w_*$ [cf. Mou22].

**Theorem 2.** *Assume that for all $j \in [d]$, $\mathrm{E}\big[\phi_j^4(X)\big] < \infty$, $\mathrm{E}\big[Y^2\big] < \infty$, and $\mathrm{E}[\|g(X,Y)\|_{\Sigma^{-1}}^2] < \infty$. Let $\delta \in (0,1)$. If*

$$n \geq (512\lambda_{\max}(V) + 6)\log(ed) + (128L + 11)\log(2/\delta),$$

*then with probability at least $1 - \delta$,*

$$\mathcal{E}(\hat{w}_n) \leq 4 \cdot (n\delta)^{-1} \cdot \mathrm{E}[\|g(X,Y)\|_{\Sigma^{-1}}^2].$$

At a high-level, this result says that above a certain explicit minimal sample size, the asymptotic expression of the excess risk of Theorem 1 is correct, up to a significantly worse dependence on $\delta$. The restriction on the sample size is almost the best one can hope for. To see why, note that to get guarantees on the excess risk of *any* empirical risk minimizer, we need at least that $\Sigma_n$ is invertible, otherwise there exists an empirical risk minimizer arbitrarily far away from $w_*$. To get quantitative guarantees, we need slightly more control in the form of a lower bound on $\lambda_{\min}(\Sigma^{-1/2}\Sigma_n\Sigma^{-1/2})$. We refer the reader to a more detailed discussion in [EME24, Section 5].

This result has two key qualities, which we aim to reproduce in our results. First, it is assumption-lean, requiring nothing more than a fourth moment assumption on the coordinates of the feature map compared to Theorem 1. Second, it recovers the right dependence on the intrinsic parameter $\mathrm{E}[\|g(X,Y)\|_{\Sigma^{-1}}^2]$ identified in Theorem 1. A downside of this generality is the bad dependence on $\delta$. Without further assumptions, this cannot be improved; we refer the reader to the recent literature on robust linear regression for more on this issue [e.g. LM19; LL20; EME24].

## 3 Main Results

In this section we state our main results. They are most easily seen as extensions of Theorems 1 and 2 for general index sets $\mathcal{T}$. In Section 3.1, we study the asymptotics of the excess risk of ERM in our setting, and in Section 3.2, we present a non-asymptotic upper bound on the excess risk.

To state our results, we require additional definitions and notation. We start with the population and the sample covariance matrices

$$\Sigma(t) := \mathrm{E}\big[\phi_t(X)\phi_t(X)^T\big], \qquad \Sigma_n(t) := n^{-1}\sum_{i=1}^n \phi_t(X_i)\phi_t(X_i)^T.$$

We define the following collection of minimizers,

$$w_*(t) := \operatorname*{argmin}_{w \in \mathbb{R}^d} R(t, w), \qquad \mathcal{T}_* := \operatorname*{argmin}_{t \in \mathcal{T}} R(t, w_*(t)),$$

the first is uniquely defined, while the second is set-valued in general. We define the gradient of the loss at these minimizers and their corresponding covariance matrices

$$g(t, (X,Y)) := \nabla_w \ell(\langle w_*(t), \phi_t(X)\rangle, Y), \qquad G(t,s) := \mathrm{E}\big[g(t, (X,Y))g(s, (X,Y))^T\big].$$

Finally, we introduce the following processes which play a key role in our development

$$\Lambda_n(t) := \sqrt{n} \cdot \lambda_{\max}(I - \Sigma^{-1/2}(t)\Sigma_n(t)\Sigma^{-1/2}(t)), \quad G_n(t) := \sqrt{n} \cdot \|\nabla_w R_n(t, w_*(t))\|_{\Sigma^{-1}(t)}, \quad (3)$$

as well as, for $t_* \in \mathcal{T}_*$ and $t \in \mathcal{T} \setminus \mathcal{T}_*$,

$$\Delta_n(t, t_*) := \sqrt{n} \cdot \left(1 - \frac{R_n(t, w_*(t)) - R_n(t_*, w_*(t_*))}{R(t, w_*(t)) - R_*}\right). \quad (4)$$

We note that the process $(\Delta_n(t, t_*))_{t \in \mathcal{T} \setminus \mathcal{T}_*}$ is an empirical process (see [VW96] for an introduction), while $(\Lambda_n(t))_{t \in \mathcal{T}}$ and $(G_n(t))_{t \in \mathcal{T}}$ are partial suprema of empirical processes. In the sequel, we will slightly abuse this terminology, and call all of these empirical processes, with the understanding that they can be viewed as one with more indexing. We will further assume that these processes are separable; see [BLM13, p.305-306] for a definition. This covers a wide range of applications, while avoiding delicate measurability issues. The suprema of such separable processes, which is the only way they enter our results, can be studied by taking the supremum over a countable dense subset of the index set. Therefore, without loss of generality, we assume that $\mathcal{T}$ is countable.

Finally, in line with the literature on the theory of empirical processes [VW96], we say that a sequence of empirical processes is Glivenko–Cantelli if, when rescaled by $n^{-1/2}$, the supremum of their absolute value taken over their index set converges to zero in probability as $n \to \infty$. In other words, the weak law of large numbers holds uniformly over the index set. Similarly, we say that a sequence of empirical processes is Donsker if it converges in distribution to its limiting Gaussian process.[3] In other words, the central limit theorem holds uniformly over the index set.

## 3.1 Asymptotic result

Our first main result is an asymptotic characterization of the quantiles of the excess risk of any sequence of empirical risk minimizers in our setting, which vastly generalizes that of Theorem 1.

**Theorem 3.** *Assume that $\mathcal{T}_* \neq \varnothing$ and for some $t_* \in \mathcal{T}_*$, assume that the empirical processes $(\Lambda_n(t))_{t \in \mathcal{T}}$, $(\Delta(t, t_*))_{t \in \mathcal{T} \setminus \mathcal{T}_*}$ and $(G_n(t))_{t \in \mathcal{T}}$ are Glivenko-Cantelli. Then, for all $\varepsilon > 0$,*

$$\lim_{n \to \infty} \mathrm{P}\big(R(\hat{t}_n, w_*(\hat{t}_n)) - R_* > \varepsilon\big) = 0.$$

*Furthermore, if the sequence of processes $(G_n(t))_{t \in \mathcal{T}}$ is Donsker, then for any $\delta \in (0, 1)$,*

$$\frac{1}{2} \cdot Q_{Z^-}(1 - \delta) \leq \liminf_{n \to \infty} n \cdot Q_{\mathcal{E}(\hat{t}_n, \hat{w}_n)}(1 - \delta) \leq \limsup_{n \to \infty} n \cdot Q_{\mathcal{E}(\hat{t}_n, \hat{w}_n)}(1 - \delta) \leq Q_{Z^+}(1 - \delta),$$

*where $Z^- := \inf_{s \in \mathcal{T}_*} \|Z(s)\|_2^2$, $Z^+ := \sup_{s \in \mathcal{T}_*} \|Z(s)\|_2^2$, and $(Z(t))_{t \in \mathcal{T}}$ is a mean-zero Gaussian process with covariance function $\mathrm{E}[Z(t)Z(s)^T] = \Sigma^{-1/2}(t)G(t, s)\Sigma^{-1/2}(s)$ for all $t, s \in \mathcal{T}$.*

We note that, up to a factor of two in the upper bound on the asymptotic quantiles, Theorem 3 reduces to Theorem 1 when $\mathcal{T}$ is a singleton, with the exact same assumptions. We are not aware of comparable results in the literature. The proof of Theorem 3 can be found in Appendix D.

*Remark* 1. For small $\delta$, the upper bound admits the more interpretable expression

$$Q_{Z_+}(1 - \delta) \asymp \mathrm{E}\left[\sup_{s \in \mathcal{T}_*} \|Z(s)\|_2^2\right] + 2\log(1/\delta) \sup_{s \in \mathcal{T}_*} \lambda_{\max}(\Sigma^{-1/2}(s)G(s, s)\Sigma^{-1/2}(s)). \tag{5}$$

Furthermore, if $\mathcal{T}_*$ is finite, the first term can be upper bounded as

$$\mathrm{E}\left[\max_{s \in \mathcal{T}_*} \|Z(s)\|_2^2\right] \leq 80 \cdot (1 + \log|\mathcal{T}_*|) \cdot \max_{s \in \mathcal{T}_*} \mathrm{E}[\|g(s, (X, Y))\|_{\Sigma^{-1}(s)}^2]. \tag{6}$$

To see why Theorem 3 is surprising, let us first focus on the case where $\mathcal{T}_*$ has a unique element $t_*$, so that $Z_+ = Z_- \overset{d}{=} \|Z\|_2^2$ where $Z \sim \mathcal{N}(0, \Sigma^{-1/2}(t_*)G(t_*, t_*)\Sigma^{-1/2}(t_*))$. Now consider the oracle procedure, which knows beforehand what the optimal feature map $t_*$ is, and outputs $t_*$ and a minimizer of $R_n(t_*, w)$. Theorem 3 says that, up to a factor of two, the asymptotic quantiles of the excess risk of ERM, which needs to learn over the large class $\cup_{t \in \mathcal{T}}\{x \mapsto \langle w, \phi_t(x) \rangle \mid w \in \mathbb{R}^d\}$, coincide with those of the oracle procedure (by Theorem 1), which only needs to learn over the *linear* class $\{x \mapsto \langle w, \phi_{t_*}(x) \rangle \mid w \in \mathbb{R}^d\}$!

More generally, Theorem 3 establishes that asymptotically, any ERM picks a near-optimal feature map with probability one. It furthers shows that the asymptotic quantiles of the excess risk of any sequence of ERMs is controlled from above and below by those of the extrema of the limiting Gaussian process of $(G_n(t))_{t \in \mathcal{T}}$ on the set of optimal feature maps $\mathcal{T}_*$. This is surprising, as it implies that asymptotically, and outside of its role in determining whether the assumptions of Theorem 3 hold, the global complexity of the set $\mathcal{T}$ is irrelevant to the excess risk of ERM.

Finally, we note that the Glivenko-Cantelli and Donsker assumptions in Theorem 3 can equivalently be viewed as restrictions on the size of $\mathcal{T}$, for distribution and process dependent notions of size. We refer the reader to the books [VW96; GN15] for more on this connection. With this observation, the main takeaway from Theorem 3 can be stated as follows.

*Asymptotically, if $\mathcal{T}$ is not too large, the excess risk of ERM depends, at worst, only on the complexity of the set of optimal feature maps $\mathcal{T}_*$, and is independent of the global complexity of $\mathcal{T}$.*

---

[3]See [VW96, Section 2.1] or the proof of Theorem 3 for a more precise definition.

## 3.2 Non-asymptotic result

The result in Theorem 3 hints at a dramatic localization phenomenon, whereby the influence of the size and complexity of the collection of feature maps $(\phi_t)_{t \in \mathcal{T}}$ on the excess risk of ERM vanishes as $n \to \infty$ under appropriate assumptions. The root of this localization phenomenon is the first statement of Theorem 3: eventually, ERM picks near-optimal feature maps with probability approaching one. For small enough sample sizes however, it is clear that ERM is likely to select suboptimal feature maps, so that this localization phenomenon cannot hold uniformly over $n$. This raises a host of questions: (i) How fast, as measured by the sample size, does ERM learn the optimal feature map? (ii) What is the effect of this localization on the rate of decay of the excess risk of ERM non-asymptotically? (iii) What properties of the feature maps $(\phi_t)_{t \in \mathcal{T}}$ influence these rates?

Our answers to these questions in this very general setting are formally expressed in Theorem 4 below. To state it, we define the following parameter

$$L := \sup \mathrm{E}\left[\left(\sum_{t \in \mathcal{T}}\langle v_t, \Sigma^{-1/2}(t)\phi_t(X)\rangle^2 - 1\right)^2\right],$$

where the supremum is taken over vectors $(v_t)_{t \in \mathcal{T}}$ such that $\sum_{t \in \mathcal{T}}\|v_t\|_2^2 = 1$. For $n \in \mathbb{N}$ and $\delta \in (0,1)$, we define the set function $F_{n,\delta}$, for any subset $\mathcal{S} \subset \mathcal{T}$, by

$$F_{n,\delta}(\mathcal{S}) := \left\{ t \in \mathcal{T} \;\middle|\; R(t, w_*(t)) - R_* \leq 2 \cdot (n\delta)^{-1} \cdot \mathrm{E}[\sup_{s \in \mathcal{S}} G_n^2(s)] \right\} \tag{7}$$

This map acts as a contraction as shown in the next lemma, whose proof is deferred to Appendix E. For a function $f$, we use $f^k$ to denote $f^k(x) := f(f^{k-1}(x))$ with $f^0(x) := x$.

**Lemma 1.** *Let $n \in \mathbb{N}$, $\delta \in (0,1)$, and assume that $\mathcal{T}_* \neq \varnothing$. Then for all $k \in \mathbb{N} \cup \{0\}$,*

- $F_{n,\delta}^{k+1}(\mathcal{T}) \subseteq F_{n,\delta}^k(\mathcal{T})$.

- *If $\exists\, n_0, B$ such that $\mathrm{E}[\sup_{t \in \mathcal{T}} G_n^2(t)] \leq B$ for all $n \geq n_0$, then $\bigcap_{n \geq 1} F_{n,\delta}^k(\mathcal{T}) = \mathcal{T}_*$.*

With these definitions, we now state the second main result of the paper. A proof is in Appendix F.

**Theorem 4.** *Assume that $\mathcal{T}_* \neq \varnothing$, $\mathrm{E}[Y^2] < \infty$, $\forall (t,j) \in \mathcal{T} \times [d]$, $\mathrm{E}[\phi_{t,j}^2(X)] < \infty$, and $\mathrm{E}[\|g(t,(X,Y))\|_{\Sigma^{-1}(t)}^2] < \infty$. Let $\delta \in (0,1)$ and $k \in \mathbb{N}$. If, for some $t_* \in \mathcal{T}_*$, $n$ satisfies*

$$n \geq 64\,\mathrm{E}[\sup_{t \in \mathcal{T}} \Lambda_n(t)] + (128L + 11)\log(6/\delta) + 6 \cdot \delta^{-2} \cdot \mathrm{E}[\sup_{t \in \mathcal{T} \setminus \mathcal{T}_*} \Delta_n(t, t_*)],$$

*then, with probability at least $1 - \delta$,*

$$\hat{t}_n \in F_{n,\delta/2k}^k(\mathcal{T}) =: \mathcal{S}_{n,\delta,k},$$

*and*

$$\mathcal{E}(\hat{t}_n, \hat{w}_n) \leq 24 \cdot (n\delta)^{-1} \cdot \mathrm{E}[\sup_{s \in \mathcal{S}_{n,\delta,k}} G_n^2(s)],$$

*where the processes $\Lambda_n$, $\Delta_n$, and $G_n$ are as in (3) and (4).*

We make a few remarks before interpreting the content of the theorem. First, we note that when the index set $\mathcal{T}$ is a singleton, the last term in the sample size restriction vanishes, while the first matches the sample size restriction from Theorem 2 after an application of Lemma 3 below; further taking $k = 1$ in Theorem 4 recovers the upper bound on the excess risk of Theorem 2 up to a constant factor. Theorem 4 may therefore be viewed as a broad generalization of Theorem 2. Second, under Assumption 1 below, and by the second item of Lemma 1, the upper bound on the excess risk in Theorem 4 eventually matches the main term in the asymptotic bound of Theorem 3 as can be seen from (5), in the same way that Theorem 2 achieves this when compared with Theorem 1. Finally, the statement of Theorem 4 is very general, and in fact, too general for us to be able to interpret it precisely. As such, we will discuss it in the context of the following assumption.

**Assumption 1.** There exists constants $C_\Lambda$, $C_\Delta$, and $C_G$ independent of the sample size, but possibly dependent on the remaining parameters of the problem, such that for all $n \in \mathbb{N}$,

$$\mathrm{E}[\sup_{t \in \mathcal{T}} \Lambda_n(t)] \leq C_\Lambda, \quad \mathrm{E}\left[\sup_{t \in \mathcal{T} \setminus \mathcal{T}_*} \Delta_n(t, t_*)\right] \leq C_\Delta, \quad \mathrm{E}[\sup_{t \in \mathcal{T}} G_n^2(t)] \leq C_G,$$

where $\Lambda_n$, $\Delta_n$, and $G_n$ are as in (3) and (4).

These assumptions can be equivalently viewed as a restriction on the appropriately measured size of the index set $\mathcal{T}$ [VW96; GN15], and are slightly stronger than the assumptions of Theorem 3. They always hold for finite index sets, and we will derive in Section 4 explicit estimates of the constants in Assumption 1 in terms of moments of the feature maps and target as well as the cardinality of $\mathcal{T}$.

Let us now interpret the content of Theorem 4, which comes with a free parameter $k$, in the context of Assumption 1. We fix $k$ here, and discuss its choice below. First, recalling the definition of $F_{n,\delta}$, this result says that above a certain sample size, both the suboptimality of the feature map picked by ERM and its excess risk decay at the fast rate $n^{-1}$, answering the first question we raised at the beginning of the section. Second, this result provides an upper bound on the excess risk of ERM that depends on the index set $\mathcal{T}$ *only* through the size of shrinking subsets $\mathcal{S}_{n,\delta,k}$, which might be large for small $n$, but which by Lemma 1 converge to the set of optimal feature maps $\mathcal{T}_*$ as $n \to \infty$. This transparently shows the effect of the localization phenomenon on the rate of decay of the excess risk of ERM, answering the second question we raised. Finally, looking at the definition of $\mathcal{S}_{n,\delta,k}$, this result identifies the size of the sublevel sets of the suboptimality function $R(t, w_*(t)) - R_*$ defined over feature maps as a relevant property of the collection of feature maps $(\phi_t)_{t \in \mathcal{T}}$ that influences the rate of convergence of the excess risk of ERM in this setting, answering the final question we raised.

Finally, let us turn to the choice of $k$. Practically, we select the one that minimizes the bound on the excess risk. Looking at the first item of Lemma 1, this optimal $k$ balances the following trade-off: on the one hand, for small $k$, applications of $F_{n,\delta/2k}$ constrain the input set more severely, but only a few iterations are performed; on the other hand, larger values of $k$ allow more iterations, but at the cost of more weakly constraining the input set per application.

Stepping back, there are two main takeaways from Theorem 4. Firstly, and on a conceptual level, it shows that feature learning is easy when the suboptimality function $R(t, w_*(t)) - R_*$, defined over the set of features maps, has small sublevel sets. Secondly, and on a technical level, it provides a template which can be used to derive more explicit excess risk bounds on ERM given estimates on the expected suprema of the relevant empirical processes. Deriving such accurate estimates for infinite $\mathcal{T}$ is a highly non-trivial task, and cannot be done at the level of generality we have been operating at. The case of finite $\mathcal{T}$ however is tractable in a general setting as we discuss in the next section.

## 4 Case study: Finite index sets

In this section, we focus on the case where the index set $\mathcal{T}$ is finite, and aim, among other things, at establishing explicit estimates on the various expected suprema appearing in Theorem 4 in terms of moments of the feature maps and of the target. This problem becomes tractable in the case of finite $\mathcal{T}$ because, roughly speaking, a worst-case analysis still yields non-trivial upper bounds. This is decidedly not the case when $\mathcal{T}$ is infinite, in which case these expected suprema can be infinite.

We start with a slight strengthening of Theorem 3, whose assumptions reduce to simple moments conditions when $\mathcal{T}$ is finite. The straightforward proof can be found in Appendix H.

**Corollary 1.** *Assume that $\mathcal{T}$ is finite, for all $(t, j) \in \mathcal{T} \times [d]$, $\mathrm{E}\big[\phi_{t,j}^2(X)\big] < \infty$, $\mathrm{E}\big[Y^2\big] < \infty$, and for all $t \in \mathcal{T}$, $\mathrm{E}[\|g(t,(X,Y))\|_{\Sigma^{-1}(t)}^2] < \infty$. Then*

$$\lim_{n \to \infty} \mathrm{P}\big(\hat{t}_n \notin \mathcal{T}_*\big) = 0.$$

*Furthermore, for any $\delta \in (0,1)$,*

$$\frac{1}{2} \cdot Q_{Z^-}(1-\delta) \leq \liminf_{n \to \infty} n \cdot Q_{\mathcal{E}(\hat{t}_n, \hat{w}_n)}(1-\delta) \leq \limsup_{n \to \infty} n \cdot Q_{\mathcal{E}(\hat{t}_n, \hat{w}_n)}(1-\delta) \leq \frac{1}{2} \cdot Q_{Z^+}(1-\delta),$$

*where $Z^- := \min_{s \in \mathcal{T}_*} \|Z_s\|_2^2$, $Z^+ := \max_{s \in \mathcal{T}_*} \|Z_s\|_2^2$, and the random vectors $(Z_t)_{t \in \mathcal{T}}$ are jointly Gaussian with mean zero and covariance $\mathrm{E}[Z_t Z_s^T] = \Sigma^{-1/2}(t) G(t,s) \Sigma^{-1/2}(s)$ for all $t, s \in \mathcal{T}$. In particular, if $\mathcal{T}_* = \{t_*\}$, then*

$$n \cdot \mathcal{E}(\hat{t}_n, \hat{w}_n) \xrightarrow{d} \frac{1}{2} \cdot \|Z\|_2^2,$$

*where $Z \sim \mathcal{N}(0, \Sigma^{-1/2}(t_*) G(t_*, t_*) \Sigma^{-1/2}(t_*))$.*

The conclusions of Corollary 1 differ from those of Theorem 3 in two aspects. First, the feature map picked by ERM is guaranteed to be optimal rather than near-optimal with probability converging to

one. Second, the upper bound on the asymptotic quantiles is improved by a factor of two, yielding the exact distribution of the rescaled excess risk when $\mathcal{T}_*$ is a singleton.

Making Theorem 4 more explicit is a more laborious task. We recall here two known results that allow us to accomplish this. We start with the following bounds on the expectation of the supremum of a finitely-indexed empirical process, which we will later use to bound the suprema of the processes $(G_n(s))_{s\in\mathcal{S}}$ and $(\Delta_n(t, t_*))_{t\in\mathcal{T}\setminus\mathcal{T}_*}$ appearing in Theorem 4. A proof can be found in Appendix G.

**Lemma 2.** *Let $n, d \in \mathbb{N}$, and let $Z$ be a random element taking value in a set $\mathcal{Z}$, and let $(Z_i)_{i=1}^n$ be i.i.d. samples with the same distribution as $Z$. Let $\mathcal{F}$ be a finite collection of $\mathbb{R}^d$-valued measurable functions. Define*

$$\sigma^2(\mathcal{F}) := \max_{f\in\mathcal{F}} \mathrm{E}\big[\|f(Z) - \mathrm{E}[f(Z)]\|_2^2\big], \qquad r_n(\mathcal{F}) := \mathrm{E}\bigg[\max_{(i,f)\in[n]\times\mathcal{F}}\|f(Z_i) - \mathrm{E}[f(Z)]\|_2^2\bigg]^{1/2},$$

*and let $E_n(f) := \sqrt{n}\cdot(n^{-1}\sum_{i=1}^n f(Z_i) - \mathrm{E}[f(Z)])$. Then, we have*

$$\frac{1}{2}\cdot\sigma(\mathcal{F}) + \frac{1}{4}\cdot\frac{r_n(\mathcal{F})}{\sqrt{n}} \leq \mathrm{E}\Big[\max_{f\in\mathcal{F}}\|E_n(f)\|_2^2\Big]^{1/2} \leq c(|\mathcal{F}|)\cdot\sigma(\mathcal{F}) + c^2(|\mathcal{F}|)\cdot\frac{r_n(\mathcal{F})}{\sqrt{n}},$$

*where $c(m) := 5\sqrt{1+\log m}$.*

Lemma 2 allows us to compute the expected supremum of a finitely-indexed empirical process, up to log factors in the size of the index set. It is known that these factors cannot be removed from the upper bound nor added to the lower bound without more assumptions, we refer the reader to a related discussion in [Tro16]. Finally, while the term $r_n(\mathcal{F})$ might grow with $n$, by bounding the maximum with the sum, it grows at most as $\sqrt{n}$. In many applications however, the random vectors $f(Z)$ are bounded almost surely, so that $r_n(\mathcal{F})$ is of order one, which justifies our presentation choice.

The second result we recall is the expectation version of a one sided Matrix Bernstein inequality due to Tropp [Tro15]. We use it below to bound the supremum of the process $(\Lambda_n(t))_{t\in\mathcal{T}}$ appearing in Theorem 4. We do not known of a matching non-asymptotic lower bound, but an asymptotic one is known [EME24, Proposition 17]. Upper and lower bounds similar to those of Lemma 2 hold if one considers the expected operator norm instead of only the maximum eigenvalue [Tro16, Section 7].

**Lemma 3** ([Tro15], Theorem 6.6.1.). *Let $n, d \in \mathbb{N}$ and for each $i \in [n]$, let $Z_i \in \mathbb{R}^{d\times d}$ be i.i.d. positive semi-definite matrices with the same distribution as $Z$. Define*

$$V := \mathrm{E}\Big[(\mathrm{E}[Z] - Z)^2\Big], \quad W_n := \sqrt{n}\cdot\Big(\mathrm{E}[Z] - \frac{1}{n}\sum_{i=1}^n Z_i\Big).$$

*Then, we have*

$$\mathrm{E}[\lambda_{\max}(W_n)] \leq \sqrt{2\lambda_{\max}(V)\log(ed)} + \frac{\lambda_{\max}(\mathrm{E}[Z])\log(ed)}{3\sqrt{n}}.$$

Equipped with these estimates, we may now control the expected suprema of the empirical processes appearing in Theorem 4. To apply Lemma 2, define the following classes, for $\mathcal{S} \subset \mathcal{T}$ and $t_* \in \mathcal{T}_*$

$$\mathcal{G}(\mathcal{S}) := \Big\{(x,y) \mapsto \Sigma^{-1/2}(s)g(s,(x,y)) \,\Big|\, s \in \mathcal{S}\Big\},$$

$$\mathcal{D}(t_*) := \Big\{(x,y) \mapsto \frac{\ell(\langle w_*(t), \phi_t(x)\rangle, y) - \ell(\langle w_*(t_*), \phi_{t_*}(x)\rangle, y)}{R(t, w_*(t)) - R_*} \,\Big|\, t \in \mathcal{T}\setminus\mathcal{T}_*\Big\}.$$

Applying Lemma 2 on $\mathcal{G}(\mathcal{S})$ bounds the expected supremum of the process $(G_n(s))_{s\in\mathcal{S}}$ while applying it on $\mathcal{D}(t_*)$ bounds that of $(\Delta_n(t, t_*))_{t\in\mathcal{T}\setminus\mathcal{T}_*}$. To control the supremum of $(\Lambda_n(t))_{t\in\mathcal{T}}$, the key idea is to notice that it can be expressed as the maximum eigenvalue of a block diagonal matrix whose blocks are $\sqrt{n}(I - \Sigma^{-1/2}(t)\Sigma_n(t)\Sigma^{-1/2}(t))$. Looking at Lemma 3, the relevant parameter is therefore a block diagonal matrix $V$ with the following blocks

$$V(t) := \mathrm{E}\bigg[\Big(\Sigma^{-1/2}(t)\phi_t(X)\phi_t(X)^T\Sigma^{-1/2}(t) - I\Big)^2\bigg].$$

As the bound in Lemma 3 depends only on the maximum eigenvalue of $V$, the ordering of the blocks does not matter. Putting together these estimates, we arrive at a fully explicit version of Theorem 4.

**Corollary 2.** *Assume that $\mathcal{T}$ is finite and that for all $(t,j) \in \mathcal{T} \times [d]$, $\mathrm{E}\big[\phi_{t,j}^4(X)\big] < \infty$, $\mathrm{E}\big[Y^4\big] < \infty$. Let $\delta \in (0,1)$, $k \in [1 + |\mathcal{T} \setminus \mathcal{T}_*|]$, and $c(\cdot), \sigma^2(\cdot), r_n(\cdot)$ as in Lemma 2. If, for some $t_* \in \mathcal{T}_*$,*

$$n \geq (512\lambda_{\max}(V) + 6)\log(ed|\mathcal{T}|) + (128L + 11)\log(6/\delta)$$
$$+ 24 \cdot \delta^{-1} \cdot c(|\mathcal{T}|)\sigma^2(\mathcal{D}(t_*)) + 10 \cdot \delta^{-1/2} \cdot c^2(|\mathcal{T}|)r_n(\mathcal{D}(t_*)), \quad (8)$$

*then, with probability at least $1 - \delta$*

$$\hat{t}_n \in \widetilde{F}_{n,\delta/2k}^k(\mathcal{T}) =: \widetilde{S}_{n,\delta,k},$$

*and*

$$\mathcal{E}(\hat{t}_n, \hat{w}_n) \leq 24 \cdot (n\delta)^{-1} \cdot A(\widetilde{S}_{n,\delta,k}),$$

*where, for $\mathcal{S} \subset \mathcal{T}$,*

$$A(\mathcal{S}) := c^2(\mathcal{S}) \cdot \Big(\sigma(\mathcal{G}(\mathcal{S})) + c(\mathcal{S}) \cdot \tfrac{r_n(\mathcal{G}(\mathcal{S}))}{\sqrt{n}}\Big)^2,$$

*and $\widetilde{F}_{n,\delta}(\mathcal{S})$ is the same as $F_{n,\delta}(\mathcal{S})$ defined in (7) but with $A(\mathcal{S})$ replacing $\mathrm{E}[\sup_{s \in \mathcal{S}} G_n^2(s)]$.*

We make a few remarks about Corollary 2; a proof sketch is in Appendix I. The set function $A(\mathcal{S})$ controlling the contraction rate of the map $\widetilde{F}_{n,\delta}$ as well as the excess risk, has a pleasantly simple form. To first order, and ignoring constants, it is given by

$$(1 + \log|\mathcal{S}|) \cdot \max_{s \in \mathcal{S}} \mathrm{E}\Big[\|g(s,(X,Y))\|_{\Sigma^{-1}(s)}^2\Big].$$

As such, as $n \to \infty$ and by Lemma 1, the upper bound on the excess risk in Corollary 2 matches the main term in the asymptotic rate derived in Theorem 3, as can be seen from (6). As the sets $\widetilde{S}_{n,\delta,k}$ are shrinking with $n$, the above expression clearly shows the decaying effect of the global complexity of $\mathcal{T}$ on the excess risk. Finally, we note that the restriction on $k$ in Corollary 2 is there only because after at most that many iterations, a fixed point is reached, and further iterations worsen the bound. We conclude this section with an example of an application of our results.

**Example 1** (Sparse linear regression). Consider the sparse linear regression problem, and in particular the best subset selection (BSS) procedure [Mil02; HTF09]. This procedure corresponds to ERM over the restricted linear class $\{x \mapsto \langle w, \phi(x) \rangle \mid \|w\|_0 \leq s\}$ in the linear regression setup of Section 2, where $\|w\|_0$ is the number of non-zero entries of $w$ and $s \in [d]$ is a user-chosen sparsity level.

The problem of computing the BSS procedure has attracted a lot of attention recently. While NP-hard and therefore difficult in the worst case [Nat95], Bertsimas et al. [BKM16] showed that it can be tractable on practical instances of moderate size. Since then, a rich literature has emerged that devises increasingly efficient methods [e.g. Hua+18; BP20; HMS22; Guy+24]. By comparison, the statistical performance of the BSS procedure is not yet completely understood as we discuss below.

To see how the sparse linear regression problem fits in our feature learning setting, notice that $\{x \mapsto \langle w, \phi(x) \rangle \mid \|w\|_0 \leq s\} = \{x \mapsto \langle v, \phi_t(x) \rangle \mid (t,v) \in \mathcal{T} \times \mathbb{R}^s\}$ where $\mathcal{T}$ is the set of all subsets of $[d]$ of size $s$, and $\phi_t(x) := (\phi_{j_1}(x), \phi_{j_2}(x), \ldots, \phi_{j_s}(x)) \in \mathbb{R}^s$ where $(j_1, j_2, \ldots, j_s)$ are the elements of $t$ in increasing order. As such, Corollaries 1 and 2 are immediately applicable and provide general statements on the performance of an arbitrary BSS procedure. To simplify the discussion, we assume for the rest of the example that there is a unique risk minimizer $w_*$ satisfying $\|w_*\|_0 = s$.

On the recovery side, the first item of Corollary 1 guarantees that we asymptotically exactly recover the support of $w_*$. Non-asymptotically, the first item of Corollary 2 shows that if $n$ further satisfies

$$n > \min_{k \in \left[\binom{d}{s}\right]} \Big\{4k \cdot (\gamma\delta)^{-1} \cdot A(\widetilde{F}_{n,\delta/2k}^{k-1}(\mathcal{T}))\Big\} \quad \text{where} \quad \gamma := \min_{t \in \mathcal{T} \setminus \mathcal{T}_*} \{R(t, w_*(t)) - R_*\},$$

then with probability at least $1 - \delta$, the BSS procedure recovers the support of $w_*$. Equivalently, these two statements say that for large enough $n$, the BSS procedure coincides with the oracle procedure which knows the support of $w_*$ a priori and outputs an ERM from the optimal linear class.

In practice however, the interesting regime is when $n$ is only moderately large. Corollary 2 provides our guarantee in this case, and as such, we turn our attention to the sample size restriction (8). Typically, we expect the main restriction to come from the first term, which in this case is given by $\lambda_{\max}(V) \cdot s\log(d/s)$, up to constants and lower order terms. This is because if an intercept is

included, i.e. $\phi_1(X) = 1$, then $\lambda_{\max}(V) \geq s - 1$, so the first term scales as $s^2 \log(d/s)$ at least, while the remaining terms typically grow more slowly with $s$. As a concrete example, when $\phi(X)$ is a Gaussian vector, $\lambda_{\max}(V) = s + 1$, so in this case the estimate $s^2 \log(d/s)$ is tight. Under this sample size restriction, and if $\varepsilon := Y - \langle w_*, \phi(X) \rangle$ satisfies $\mathrm{E}\big[\varepsilon^2 \mid X\big] \leq \sigma^2$, Corollary 2 upper bounds the excess risk by $(\sigma^2 s/n) \cdot a_n$ for a sequence of decreasing distribution-dependent constants $a_n$ converging to one as $n \to \infty$, ignoring the dependence on $\delta$ and absolute constants.

The closest existing result in the literature we are aware of is due to Shen et al. [She+13], who arrived at comparable conclusions but in a substantially different setting. In particular, their result was obtained in the setting $n, d \to \infty$, with an implicit assumption on the distribution of $\phi(X)$ [She+13, Equation 2], and dealt with the in-sample prediction risk instead of the excess risk. Another closely related result is due to Raskutti et al. [RWY11] who showed that the minimax expected excess risk in a well-specified fixed-design setting is, up to constants, $\sigma^2 s \log(d/s)/n$; see also [Bac24, Chapter 8]. Our results show that for moderate $n$, in the random-design setting, and when focusing on a single instance, the $\log(d/s)$ factor can be replaced with another factor that decays to one as $n \to \infty$.

Coming back to the sample size restriction discussion, we strongly suspect that the factor $s \log(d/s)$ is suboptimal, but we are unsure what the correct dependence is, even under Gaussian $\phi(X)$. Indeed, this factor comes from the logarithmic factor in Lemma 3, when applied to the block diagonal matrix with blocks $\sqrt{n}\big(I - \Sigma^{-1/2}(t)\Sigma_n(t)\Sigma^{-1/2}(t)\big)$. One can improve this factor by instead using versions of this inequality based on the intrinsic dimension [Tro15, Chapter 7]. However, this is also unlikely to be tight. Roughly speaking, this is because such logarithmic factors are tight only when the eigenvalues of the random matrix are near-independent. This is certainly not the case for the block diagonal matrix we are considering, since its blocks are sample covariance matrices of sub-vectors of the same random vector $\phi(X)$. Capturing this dependence is beyond our reach and likely requires new tools; we refer the interested reader to the recent articles [vHan17; LvHY18; BBvH23].

## 5 Conclusion

Broadly speaking, there are two main conclusions one can draw from this work. Firstly, in the large sample regime, and if the set of candidate feature maps is not too large under an appropriate measure of size, asking a model to additionally pick a feature map on top of learning a linear predictor has a negligible effect on the excess risk of ERM on regression problems with square loss. Secondly, for moderate sample sizes, the magnitude of this effect depends on the appropriately measured size of the sublevel sets of the suboptimality function $t \mapsto R(t, w_*(t)) - R_*$. Plainly, learning feature maps is easy when only a small subset of them is good, as the bad ones can be quickly discarded.

The most tantalizing aspect of our results is their potential in explaining the experiments in [Zha+21]. It was shown there that complex neural networks trained by ERM were able to achieve good performance despite being expressive enough to fit random labels. This is paradoxical if one assumes that the performance of ERM is driven by the complexity of the model class. Our results refute this assumption for a generic collection of feature-learning-based models. While there are many works offering explanations for this apparent paradox (see e.g. [BMR21] for a survey), we are not aware of one that shows the vanishing influence of the size of the model class on the excess risk as Theorems 3 and 4 show. Formally connecting our statements to these experiments is beyond what we achieved here, yet, we believe that the new perspective we took might generate useful insights in this area.

We conclude by outlining a few limitations of our work. Firstly, we do not deal with the question of how to solve the ERM problem. Our focus is on understanding its statistical performance, and our setting is so general that such a question cannot be meaningfully tackled. Continuing on this last point, while the generality of our results is desirable in some aspects, it is detrimental in others. As an example, it would be desirable to specialize our results from Section 3 to specific infinite collections of feature maps used in practice. Let us also mention that it is a priori unclear whether ERM is an optimal procedure, in a minimax sense, for the model classes we consider; we suspect that recently developed tools might be relevant to address this question [Mou22]. Finally, while we focused on the case of regression with square loss, this was mostly done to simplify the presentation. Indeed, the only property of the loss used in the proofs is the exactness of its second order Taylor expansion. This is however not required if one can control the error term from above and below. It is known how to do this for many loss functions [e.g. OB21; EE23], and most importantly for logistic regression [Bac10; Bac14]. We have purposefully selected generic notation to make translating such arguments easier.

## Acknowledgments and Disclosure of Funding

Resources used in preparing this research were provided in part by the Province of Ontario, the Government of Canada through CIFAR, and companies sponsoring the Vector Institute. CM acknowledges the support of the Natural Sciences and Engineering Research Council of Canada (NSERC), RGPIN-2021-03445. MAE was partially supported by NSERC Grant [2019-06167], CIFAR AI Chairs program, and CIFAR AI Catalyst grant.

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

# A    Proof of Theorem 1

Let $A_n$ denote the event that $\Sigma_n$ is invertible. By the weak law of large numbers, $\Sigma_n$ converges to $\Sigma$ in probability so that $\lim_{n\to\infty} \mathrm{P}(A_n^c) = 0$. Now on the event $A_n$, we have by (1)

$$\sqrt{n} \cdot (\hat{w}_n - w_*) = \Sigma_n^{-1} \cdot (\sqrt{n} \cdot \nabla R_n(w_*)).$$

By the continuous mapping theorem, $\Sigma_n^{-1}$ converges to $\Sigma^{-1}$ in probability and by the central limit theorem

$$\sqrt{n} \cdot \nabla R_n(w_*) \xrightarrow{d} \mathcal{N}(0, G).$$

Therefore, by Slutsky's theorem

$$\sqrt{n} \cdot (\hat{w}_n - w_*) \xrightarrow{d} \mathcal{N}(0, \Sigma^{-1} G \Sigma^{-1}).$$

Now since the risk is quadratic and the gradient vanishes at $w_*$,

$$n \cdot [R(\hat{w}) - R(w_*)] = \frac{1}{2} \cdot \|\sqrt{n} \cdot (\hat{w} - w_*)\|_\Sigma^2 \xrightarrow{d} \frac{1}{2} \|Z\|_2^2,$$

where $Z$ is as in the theorem, and where the last statement follows by the continuous mapping theorem. This proves the first statement. The bounds on the quantiles are a consequence of concentration bounds for the norm of Gaussian vectors [e.g. EME24, Corollary 33].

# B    Proof of Theorem 2

Denote by $A_n$ the event that

$$\lambda_{\min}\left(\Sigma^{-1/2}\Sigma_n\Sigma^{-1/2}\right) \geq \frac{1}{2}.$$

We show that under the sample size restriction, $\mathrm{P}(A_n) \geq 1 - \delta/2$. Indeed we have the variational representation

$$\lambda_{\max}(I - \Sigma^{-1/2}\Sigma_n\Sigma^{-1/2}) = \sup_{v \in S^{d-1}} \frac{1}{n} \sum_{i=1}^{n} 1 - \langle v, \Sigma^{-1/2}\phi(X_i)\rangle^2.$$

Each element in the sum is upper bounded by 1, and the variance parameter in Bousquet's concentration inequality [Bou02] is given by the parameter $L$ in the statement of the theorem. Applying this inequality yields that with probability at least $1 - \delta/2$

$$\lambda_{\max}(I - \Sigma^{-1/2}\Sigma_n\Sigma^{-1/2}) \leq 2\,\mathrm{E}\left[\lambda_{\max}(I - \Sigma^{-1/2}\Sigma_n\Sigma^{-1/2})\right] + \sqrt{\frac{2L\log(2/\delta)}{n}} + \frac{4\log(2/\delta)}{3n}.$$

Using Lemma 3 to upper bound the above expectation, and replacing the sample size $n$ in the resulting inequality with the minimal allowed by the theorem proves that $\mathrm{P}(A_n) \geq 1 - \delta/2$ for all sample sizes allowable by the theorem. Now on this event we have, using (2),

$$R(\hat{w}_n) - R(w_*) = \frac{1}{2} \cdot \|\Sigma_n^{-1}\nabla R_n(w_*)\|_\Sigma^2 \leq 2 \cdot \|\nabla R_n(w_*)\|_{\Sigma^{-1}}^2.$$

An elementary calculation shows

$$\mathrm{E}\left[\|\nabla R_n(w_*)\|_{\Sigma^{-1}}^2\right] = n^{-1}\,\mathrm{E}\left[\|g(X,Y)\|_{\Sigma^{-1}}^2\right],$$

so that an application of Markov's inequality yields that there is an event $B_n$ that holds with probability at least $1 - \delta/2$ and on which

$$\|\nabla R_n(w_*)\|_{\Sigma^{-1}}^2 \leq 2 \cdot (n\delta)^{-1} \cdot \mathrm{E}\left[\|g(X,Y)\|_{\Sigma^{-1}}^2\right].$$

The union bound $\mathrm{P}(A_n \cap B_n) = 1 - \mathrm{P}(A_n \cup B_n) \geq 1 - \delta$ finishes the proof.

## C  Main Lemma

We state here a core lemma, which we use in many of our proofs. To state it, we define, for a function $F : \mathcal{S} \to \mathbb{R}$ on a subset $\mathcal{S} \subseteq \mathcal{T}$,

$$\|F\|_\infty := \sup_{s \in \mathcal{S}} |F(s)|, \quad \|F\|_{\infty,-} := \sup_{s \in \mathcal{S}} \{-F(s)\}, \quad \|F\|_{\infty,+} := \sup_{s \in \mathcal{S}} F(s),$$

where the first quantity is the $\ell^\infty$ norm of the function $F$, and the remaining are one-sided variants of it. The processes appearing in the next statement are defined in (3) and (4).

**Lemma 4.** *Assume that $\mathcal{T}_* \neq \varnothing$ and let $t_* \in \mathcal{T}_*$. On the event that $\|n^{-1/2}\Delta_n(\cdot, t_*)\|_{\infty,+} < 1$ and $\|n^{-1/2}\Lambda_n\|_{\infty,+} < 1$, we have*

$$R(\hat{t}_n, w_*(\hat{t}_n)) - R_* \leq \frac{1}{2} \cdot \frac{1}{1 - \|n^{-1/2}\Delta_n(\cdot, t_*)\|_{\infty,+}} \cdot \frac{1}{1 - \|n^{-1/2}\Lambda_n\|_{\infty,+}} \cdot (n^{-1} G_n^2(\hat{t}_n)),$$

*and*

$$\frac{1}{2} \cdot \frac{n^{-1} G_n^2(\hat{t}_n)}{(1 + \|n^{-1/2}\Lambda_n\|_{\infty,-})^2} \leq R(\hat{t}_n, \hat{w}_n) - R(\hat{t}_n, w_*(\hat{t}_n)) \leq \frac{1}{2} \cdot \frac{n^{-1} G_n^2(\hat{t}_n)}{(1 - \|n^{-1/2}\Lambda_n\|_{\infty,+})^2}.$$

*Proof.* To lighten the notation, we drop the dependence on $n$, and write $\hat{t}$ instead of $\hat{t}_n$. We start with the first statement. First, we note that if $\hat{t} \in \mathcal{T}_*$, then the statement holds trivially as the left-hand side is zero, so we only consider the other case in what follows. For any $t \in \mathcal{T}$, define

$$\hat{w}(t) \in \operatorname*{argmin}_{w \in \mathbb{R}^d} R_n(t, w),$$

where the choice of minimizer is arbitrary. With this definition, we have $\hat{w}_n = \hat{w}(\hat{t})$. Now, by definition of ERM,

$$R_n(\hat{t}, \hat{w}(\hat{t})) - R_n(t_*, w_*(t_*)) \leq 0. \tag{9}$$

On the other hand, for any $t \in \mathcal{T} \setminus \mathcal{T}_*$, we have the decomposition

$$R_n(t, \hat{w}(t)) - R_n(t_*, w_*) = [R_n(t, \hat{w}(t)) - R_n(t, w_*(t))] + [R_n(t, w_*(t)) - R_n(t_*, w_*(t_*))]. \tag{10}$$

We study each of the terms of (10) separately, and we start with the first. Note that since we are in the event

$$\inf_{t \in \mathcal{T}} \lambda_{\min}(\Sigma^{-1/2}(t) \Sigma_n(t) \Sigma^{-1/2}(t)) = 1 - \|n^{-1/2}\Lambda_n\|_{\infty,+} > 0,$$

the sample covariance matrices $\Sigma_n(t)$ are invertible for all $t \in \mathcal{T}$, so that $\hat{w}(t)$ is uniquely defined and satisfies

$$\hat{w}(t) = w_*(t) - \Sigma_n^{-1}(t) \nabla_w R_n(t, w_*(t)). \tag{11}$$

Furthermore, since the function $w \mapsto R_n(t, w)$ is quadratic in $w$ and its gradient vanishes at its minimizer $\hat{w}(t)$, we have

$$R_n(t, \hat{w}(t)) - R_n(t, w_*(t)) = -\frac{1}{2} \|\hat{w}(t) - w_*(t)\|_{\Sigma_n(t)}^2 = -\frac{1}{2} \|\nabla_w R_n(t, w_*(t))\|_{\Sigma_n^{-1}(t)}^2, \tag{12}$$

where the last equality follows from (11). To bound this last term, define

$$\widetilde{\Sigma}_n(t) := \Sigma^{-1/2}(t) \Sigma_n(t) \Sigma^{-1/2}(t).$$

Then we have,

$$\begin{aligned}
\|\nabla_w R_n(t, w_*(t))\|_{\Sigma_n^{-1}(t)}^2 &= \left\{\Sigma^{-1/2}(t) \nabla_w R_n(t, w_*(t))\right\}^T \widetilde{\Sigma}_n^{-1}(t) \left\{\Sigma^{-1/2}(t) \nabla_w R_n(t, w_*(t))\right\} \\
&\leq \lambda_{\max}(\widetilde{\Sigma}_n^{-1}(t)) \cdot \|\nabla_w R_n(t, w_*(t))\|_{\Sigma^{-1}(t)}^2 \\
&= \frac{1}{1 - \lambda_{\max}(I - \widetilde{\Sigma}_n(t))} \cdot (n^{-1} G_n^2(t)) \\
&\leq \frac{1}{1 - \|n^{-1/2}\Lambda_n\|_{\infty,+}} \cdot (n^{-1} G_n^2(t)). \tag{13}
\end{aligned}$$

Finally, the second term of (10) is lower bounded by

$$
\begin{aligned}
R_n(t, w_*(t)) - R_n(t_*, w_*(t_*))) &= (1 - n^{-1/2}\Delta_n(t, t_*))[R(t, w_*(t)) - R_*] \\
&\geq (1 - \|n^{-1/2}\Delta_n(\cdot, t_*)\|_{\infty, +})[R(t, w_*(t)) - R_*] \quad (14)
\end{aligned}
$$

Combining (13) and (12) lower bounds the first term of (10), while (14) lower bounds the second. Combining the resulting lower bound on (10) with (9) and rearranging yields the first statement.

For the upper bound in the second statement, note that for all $t \in \mathcal{T}$,

$$
\begin{aligned}
R(t, \hat{w}(t)) - R(t, w_*(t)) &= \frac{1}{2} \cdot \|\hat{w}(t) - w_*(t)\|_{\Sigma(t)}^2 \\
&= \frac{1}{2} \cdot \|\Sigma_n^{-1}(t)\nabla_w R_n(t, w_*(t))\|_{\Sigma(t)}^2 \\
&\leq \frac{1}{2} \cdot \lambda_{\max}(\widetilde{\Sigma}_n^{-2}(t)) \cdot \|\nabla_w R_n(t, w_*(t))\|_{\Sigma^{-1}(t)}^2 \\
&= \frac{1}{2} \cdot \frac{1}{(1 - \|n^{-1/2}\Lambda_n\|_{\infty, +})^2} \cdot (n^{-1}G_n^2(t)).
\end{aligned}
$$

where the second line follows from (11). In particular the inequality holds for $\hat{t}$. The lower bound holds by a similar argument.

$\square$

## D   Proof of Theorem 3

**Consistency of $\hat{t}_n$.** We want to show that, as $n \to \infty$,

$$
R(\hat{t}_n, w_*(\hat{t}_n)) - R_* \xrightarrow{p} 0. \quad (15)
$$

Using the notation introduced in Appendix C, the Glivenko-Cantelli assumptions in Theorem 3 amount to the statements that, for some $t_* \in \mathcal{T}_*$, and as $n \to \infty$,

$$
\|n^{-1/2}\Lambda_n\|_{\infty} \xrightarrow{p} 0, \quad \|n^{-1/2}\Delta_n(\cdot, t_*)\|_{\infty} \xrightarrow{p} 0, \quad \|n^{-1/2}G_n\|_{\infty} \xrightarrow{p} 0. \quad (16)
$$

Let $A_n$ denote the event that both $\|n^{-1/2}\Lambda_n\|_{\infty} < 1$ and $\|n^{-1/2}\Delta_n(\cdot, t_*)\|_{\infty} < 1$. The union bound and (16) show that

$$
\lim_{n \to \infty} \mathrm{P}(A_n^c) = 0
$$

Furthermore, on the event $A_n$, the first bound of Lemma 4 holds, and bounding $n^{-1}G_n^2(\hat{t}_n)$ by $\|n^{-1/2}G_n\|_{\infty}^2$ yields that on $A_n$

$$
R(\hat{t}_n, w_*(\hat{t}_n)) - R_* \leq \frac{1}{2} \cdot \frac{1}{1 - \|n^{-1/2}\Delta_n(\cdot, t_*)\|_{\infty}} \cdot \frac{1}{1 - \|n^{-1/2}\Lambda_n\|_{\infty}} \cdot \|n^{-1/2}G_n\|_{\infty}^2 \quad (17)
$$

Now let $\varepsilon > 0$, and denote by $B_n(\varepsilon)$ the event that the right hand side of (17) is strictly larger than $\varepsilon$. Then the statements (16) together with the continuous mapping theorem show that $\lim_{n \to \infty} \mathrm{P}(B_n(\varepsilon)) = 0$. Therefore, again by (17), we have

$$
\mathrm{P}\big(R(\hat{t}_n, w_*(\hat{t}_n)) - R_* > \varepsilon\big) \leq \mathrm{P}(B_n(\varepsilon)) + \mathrm{P}(A_n^c),
$$

and taking $n \to \infty$ proves (15).

**Asymptotic quantiles.** We start with the upper bound. We have the simple decomposition

$$
n \cdot \big[R(\hat{t}_n, \hat{w}_n) - R_*\big] = n\big[R(\hat{t}_n, \hat{w}_n) - R(\hat{t}_n, w_*(\hat{t}_n))\big] + n\big[R(\hat{t}_n, w_*(\hat{t}_n)) - R_*\big]. \quad (18)
$$

Now on the event $A_n$ defined above, we have, by an application of Lemma 4, combining the two bounds in the lemma along with (18), that the rescaled excess risk is upper bounded by

$$
\frac{1}{2} \cdot \frac{1}{1 - \|n^{-1/2}\Lambda_n\|_{\infty}} \cdot \left( \frac{1}{1 - \|n^{-1/2}\Delta_n(\cdot, t_*)\|_{\infty}} + \frac{1}{1 - \|n^{-1/2}\Lambda_n\|_{\infty}} \right) \cdot G_n^2(\hat{t}_n). \quad (19)
$$

From the Glivenko-Cantelli assumptions (16), the first three factors converge in probability to 1. Our aim will be to bound the upper tail of the last factor, which will imply a bound on the upper tail of the rescaled excess risk.

We briefly make explicit the Donsker assumption before deriving this bound. Both define and note

$$G_n(t,v) := \sqrt{n} \cdot \langle v, \Sigma^{-1/2}(t) \nabla_w R_n(t, w_*(t)) \rangle, \qquad G_n(t) = \sup_{v \in S^{d-1}} G_n(t,v),$$

where $S^{d-1}$ is the Euclidean unit sphere in $\mathbb{R}^d$. As pointed out in Section 3, the processes $G_n(t)$ are partial suprema of the empirical processes $G_n(t,v)$. The Donsker assumption of the theorem states that the empirical processes $G_n(t,v)$ take value in the space of bounded functions on $\mathcal{T} \times S^{d-1}$, equipped with the $\ell^\infty(\mathcal{T} \times S^{d-1})$ norm and the metric it induces, and converge weakly to their unique Gaussian limit $(G(t,v))_{(t,v) \in \mathcal{T} \times S^{d-1}}$ as $n \to \infty$. By inspecting their finite dimensional distributions, it is straightforward to verify that $(G(t,v))_{(t,v) \in \mathcal{T} \times S^{d-1}} \overset{d}{=} (\langle v, Z(t) \rangle)_{(t,v) \in \mathcal{T} \times S^{d-1}}$ where $(Z(t))_{t \in \mathcal{T}}$ is the $\mathbb{R}^d$-valued Gaussian process defined in the statement of Theorem 3. Finally, we define $G(t) := \sup_{v \in S^{d-1}} G(t,v)$ in analogy with the definition of $G_n(t)$.

We now upper bound the upper tail of $G_n^2(\hat{t}_n)$ in (19). Let $(\varepsilon_k)_{k=1}^\infty$ be a decreasing sequence of positive numbers such that $\varepsilon_k \to 0$ as $k \to \infty$, and define the sets

$$\mathcal{T}_*(\varepsilon) := \{ t \in \mathcal{T} \mid R(t, w_*(t)) - R_* \le \varepsilon \} \tag{20}$$

as well as the function $F_k : \ell^\infty(\mathcal{T} \times S^{d-1}) \to \mathbb{R}$ by

$$F_k(z) := \sup_{s \in \mathcal{T}_*(\varepsilon_k)} \sup_{v \in S^{d-1}} z(s,v).$$

Note on the one hand that $\cap_{k \ge 1} \mathcal{T}_*(\varepsilon_k) = \mathcal{T}_*$, and on the other that $F_k$ is continuous for all $k \in \mathbb{N}$, and in fact Lipschitz. Indeed, let $z, z' \in \ell^\infty(\mathcal{T} \times S^{d-1})$. Then

$$|F_k(z) - F_k(z')| = \left| \sup_{s \in \mathcal{T}_*(\varepsilon_k)} \sup_{v \in \mathbb{S}^{d-1}} z(s,v) - \sup_{s \in \mathcal{T}_*(\varepsilon_k)} \sup_{v \in \mathbb{S}^{d-1}} z'(s,v) \right| \le \|z - z'\|_\infty$$

Now let $k \in \mathbb{N}$ and $x \in [0, \infty)$. Then

$$\begin{aligned}
\mathrm{P}\big(G_n^2(\hat{t}_n) > x\big) &= \mathrm{P}\big(\{G_n^2(\hat{t}_n) > x\} \cap \{\hat{t}_n \in \mathcal{T}_*(\varepsilon_k)\}\big) + \mathrm{P}\big(\{G_n^2(\hat{t}_n) > x\} \cap \{\hat{t}_n \notin \mathcal{T}_*(\varepsilon_k)\}\big) \\
&\le \mathrm{P}\big(\{G_n^2(\hat{t}_n) > x\} \cap \{\hat{t}_n \in \mathcal{T}_*(\varepsilon_k)\}\big) + \mathrm{P}\big(\{\hat{t}_n \notin \mathcal{T}_*(\varepsilon_k)\}\big) \\
&\le \mathrm{P}\left( \sup_{s \in \mathcal{T}_*(\varepsilon_k)} G_n^2(s) > x \right) + \mathrm{P}\big(R(\hat{t}_n, w_*(\hat{t}_n)) - R_* > \varepsilon_k\big) \\
&= \mathrm{P}\big(F_k^2(G_n) > x\big) + \mathrm{P}\big(R(\hat{t}_n, w_*(\hat{t}_n)) - R_* > \varepsilon_k\big)
\end{aligned}$$

taking the limit as $n \to \infty$, the first term converges, by the continuous mapping theorem, to the probability of the event $\{F_k^2(G) > x\}$, where $G$ is the limiting Gaussian process discussed above, while the second term vanishes by the first part of Theorem 3. Therefore, for all $k \in \mathbb{N}$,

$$\limsup_{n \to \infty} \mathrm{P}\big(G_n^2(\hat{t}_n) > x\big) \le \mathrm{P}\left( \sup_{s \in \mathcal{T}_*(\varepsilon_k)} G^2(s) > x \right)$$

Taking the limit as $k \to \infty$, noticing that the events

$$\left\{ \sup_{s \in \mathcal{T}_*(\varepsilon_k)} G^2(s) > x \right\}$$

are nested, using the continuity of probability from above, and recalling that $\cap_{k \ge 1} \mathcal{T}_*(\varepsilon_k) = \mathcal{T}_*$ gives

$$\limsup_{n \to \infty} \mathrm{P}\big(G_n^2(\hat{t}_n) > x\big) \le \mathrm{P}\left( \sup_{s \in \mathcal{T}_*} G^2(s) > x \right).$$

Using properties of the quantile function (e.g. [EME24, Lemma 20]) finishes the proof of the upper bound. For the lower bound, we make a similar argument. We have, by an application of Lemma 4,

$$\begin{aligned}
n \cdot [R(\hat{t}_n, \hat{w}_n) - R_*] &\ge n \cdot [R(\hat{t}_n, \hat{w}_n) - R(\hat{t}_n, w_*(\hat{t}_n))] \\
&\ge \frac{1}{2} \cdot \frac{1}{(1 + \|n^{-1/2} \Lambda_n\|_{\infty,-})^2} \cdot G_n^2(\hat{t}_n).
\end{aligned}$$

By the Glivenko-Cantelli assumption on $\Lambda_n$, the first two factors converge to $1/2$. For the third, we will lower bound its upper tails, which will imply a lower bound on the upper tails of the rescaled excess risk. We let $(\varepsilon_k)_{k=1}^\infty$ be a decreasing sequence of positive numbers such that $\varepsilon_k \to 0$ as $k \to \infty$, and define $H_k : \ell^\infty(\mathcal{T} \times S^{d-1}) \to \mathbb{R}$ by

$$H_k(z) := \inf_{t \in \mathcal{T}_*(\varepsilon_k)} \sup_{v \in S^{d-1}} z(t, v),$$

where the subsets $\mathcal{T}_*(\varepsilon_k)$ are as defined in (20). Clearly, for $z, z' \in \ell^\infty(\mathcal{T} \times S^{d-1})$,

$$|H_k(z) - H_k(z')| \leq \|z - z'\|_\infty$$

so $H_k$ is Lipschitz and therefore continuous. Now

$$\mathrm{P}\big(G_n^2(\hat{t}_n) > x\big) \geq \mathrm{P}\big(\{G_n^2(\hat{t}_n) > x\} \cap \{\hat{t}_n \in \mathcal{T}_*(\varepsilon_k)\}\big)$$

$$\geq \mathrm{P}\bigg(\inf_{s \in \mathcal{T}_*(\varepsilon_k)} G_n^2(s) > x\bigg) - \mathrm{P}\big(\{\hat{t}_n \notin \mathcal{T}_*(\varepsilon_k)\}\big)$$

$$= \mathrm{P}\big(H_k^2(G_n) > x\big) - \mathrm{P}\big(R(\hat{t}_n, w_*(\hat{t}_n)) - R_* > \varepsilon_k\big)$$

By the same argument as above, we obtain, as $n \to \infty$, and for all $k \in \mathbb{N}$,

$$\liminf_{n \to \infty} \mathrm{P}\big(G_n^2(\hat{t}_n) > x\big) \geq \mathrm{P}\bigg(\inf_{s \in \mathcal{T}_*(\varepsilon_k)} G^2(s) > x\bigg)$$

Taking the limit as $k \to \infty$, and noticing that

$$\bigcup_{k \geq 1}\bigg\{\inf_{s \in \mathcal{T}_*(\varepsilon_k)} G^2(s) > x\bigg\} = \bigg\{\inf_{s \in \mathcal{T}_*} G^2(s) > x\bigg\}$$

proves that

$$\liminf_{n \to \infty} \mathrm{P}\big(G_n^2(\hat{t}_n) > x\big) \geq \mathrm{P}\bigg(\inf_{s \in \mathcal{T}_*} G^2(s) > x\bigg).$$

Using properties of the quantile function (e.g. [EME24, Lemma 20]) finishes the proof of the lower bound. The estimates on the quantiles of $Z^+$ in Remark 1 are a consequence of standard Gaussian concentration, see [e.g. EME24, Appendix A.3]. Finally, for the second statement in Remark 1,

$$\mathrm{E}\bigg[\max_{s \in \mathcal{T}_*}\|Z(s)\|_2^2\bigg] \leq \mathrm{E}\Bigg[\bigg(\sum_{s \in \mathcal{T}_*}\|Z(s)\|_2^{2p}\bigg)^{1/p}\Bigg]$$

$$\leq \bigg(\sum_{s \in \mathcal{T}_*} \mathrm{E}\big[\|Z(s)\|_2^{2p}\big]\bigg)^{1/p}$$

$$\leq 32 \cdot p \cdot \bigg(\sum_{s \in \mathcal{T}_*} \mathrm{E}\big[\|Z(s)\|_2^2\big]^p\bigg)^{1/p},$$

where the last estimate follows from Gaussian concentration. Taking $p = 1 + \log|\mathcal{T}_*|$, and recalling that for $x \in \mathbb{R}^d$, $\|x\|_p \leq d^{1/p}\|x\|_\infty$ yields the result.

## E  Proof of Lemma 1

We prove the first statement by induction. For $k = 0$, this follows directly from the fact that by definition $F_{n,\delta}^0(\mathcal{T}) = \mathcal{T}$ and $F_{n,\delta}(\mathcal{T}) \subseteq \mathcal{T}$. Now let $k \in \mathbb{N}$ and assume that the statement holds for $k-1$. Let $s \in F_{n,\delta}^{k+1}(\mathcal{T})$. Then by definition

$$R(s, w_*(s)) - R_* \leq 2 \cdot (n\delta)^{-1} \cdot \mathrm{E}[\sup_{s \in F_{n,\delta}^k(\mathcal{T})} G_n^2(s)] \leq 2 \cdot (n\delta)^{-1} \cdot \mathrm{E}[\sup_{s \in F_{n,\delta}^{k-1}(\mathcal{T})} G_n^2(s)].$$

where the second inequality follows from the fact that by the induction hypothesis, $F_{n,\delta}^k(\mathcal{T}) \subseteq F_{n,\delta}^{k-1}(\mathcal{T})$, and that the supremum is increasing. Therefore $s \in F_{n,\delta}^k(\mathcal{T})$ since the last inequality is the defining inequality for $F_{n,\delta}^k(\mathcal{T})$. We now turn to the second statement. Fix $k$ and $\delta$. On the one hand, $\mathcal{T}_* \subseteq \bigcap_{n \geq 1} F_{n,\delta}^k(\mathcal{T})$. On the other, for any $t \in \bigcap_{n \geq 1} F_{n,\delta}^k(\mathcal{T})$, we have for all $n \geq n_0$, $R(t, w_*(t)) - R_* \leq 2B \cdot (n\delta)^{-1}$. Therefore $R(t, w_*(t)) - R_* = 0$, and hence $t \in \mathcal{T}_*$.

# F  Proof of Theorem 4

Recall the notation introduced in Appendix C. Let $A_n(t_*)$ be the event that:

$$\|n^{-1/2}\Lambda_n\|_{\infty,+} \le 1/2, \quad \text{and} \quad \|n^{-1/2}\Delta_n(\cdot, t_*)\| \le 1/2.$$

We start by showing that under the sample size inequality stated in the theorem, there exists a $t_* \in \mathcal{T}_*$ such that $\mathrm{P}(A_n(t_*)) \ge 1 - \delta/3$. Indeed, we have

$$\|n^{-1/2}\Lambda_n\|_{\infty,+} = \sup_{(t,v)\in(\mathcal{T}\times S^{d-1})} \frac{1}{n}\sum_{i=1}^{n}\Big\{1 - \langle v, \Sigma^{-1/2}(t)\phi_t(X_i)\rangle^2\Big\}$$

The elements of this sum are bounded by 1, and the variance parameter of Bousquet's inequality [Bou02] is given by $L$ as defined in Section 3.2. Applying this inequality yields that with probability at least $1 - \delta/6$

$$\|n^{-1/2}\Lambda_n\|_{\infty,+} \le \frac{2}{n^{1/2}} \cdot \mathrm{E}\Big[\sup_{t\in\mathcal{T}}\Lambda_n(t)\Big] + \sqrt{\frac{2L\log(6/\delta)}{n}} + \frac{4\log(6/\delta)}{3n}$$

Furthermore, by Markov's inequality, with probability at least $1 - \delta/6$

$$\|n^{-1/2}\Delta(\cdot, t_*)\|_{\infty,+} \le \frac{6 \cdot \mathrm{E}[\sup_{t\in\mathcal{T}\setminus\mathcal{T}_*}\Delta(t, t_*)]}{n^{1/2}\cdot\delta}$$

Hence, when the inequality on the sample size stated in the theorem holds for some $t_*$, the event $A_n(t_*)$ holds with probability at least $1 - \delta/3$. Now on this event, the first bound of Lemma 4 applies, and we have

$$R(\hat{t}_n, w_*(\hat{t}_n)) - R_* \le 2 \cdot n^{-1} \cdot G_n^2(\hat{t}_n). \tag{21}$$

Now we use the iterative localization method of Koltchinskii [Kol06]. Initially, we have no information about where $\hat{t}_n$ is located aside from belonging to $\mathcal{T}$, so we start with the bound

$$R(\hat{t}_n, w_*(\hat{t}_n)) - R_* \le 2 \cdot n^{-1} \cdot \sup_{t\in\mathcal{T}} G_n^2(t). \tag{22}$$

Using Markov's inequality, we have on an event $B_{n,1}$ which holds with probability at least $1 - \delta/2k$

$$\sup_{t\in\mathcal{T}} G_n^2(t) \le 2k \cdot \delta^{-1} \cdot \mathrm{E}[\sup_{t\in\mathcal{T}} G_n^2(t)].$$

Replacing in (22) yields that on the event $A_n(t_*) \cap B_{n,1}$,

$$R(\hat{t}_n, w_*(\hat{t}_n)) - R_* \le 4k \cdot (n\delta)^{-1} \cdot \mathrm{E}[\sup_{t\in\mathcal{T}} G_n^2(t)],$$

which shows that on this event, $\hat{t}_n \in F_{n,\delta/2k}(\mathcal{T})$, by definition of the map $F_{n,\delta/2k}$. With this knowledge, we now reuse the bound (21) to obtain that on $A_n(t_*) \cap B_{n,1}$

$$R(\hat{t}_n, w_*(\hat{t}_n)) - R_* \le 2 \cdot n^{-1} \cdot \sup_{t\in F_{n,\delta/2k}(\mathcal{T})} G_n^2(t).$$

Iterating the procedure we just described $k$ times, we obtain that on an event $A_n(t_*) \cap (\cap_{j=1}^{k} B_{n,j})$, where $\mathrm{P}(B_{n,j}) \ge 1 - \delta/2k$ for all $j \in [k]$

$$\hat{t}_n \in F_{n,\delta/2k}^k(\mathcal{T}) = \mathcal{S}_{n,\delta,k}. \tag{23}$$

Another application of Markov's inequality yields that on an event $C$ which holds with probability at least $1 - \delta/6$

$$\sup_{t\in\mathcal{S}_{n,\delta,k}} G_n^2(t) \le 6 \cdot \delta^{-1} \cdot \mathrm{E}[\sup_{t\in\mathcal{S}_{n,\delta,k}} G_n^2(t)] \tag{24}$$

Since

$$\mathrm{P}(A_n(t_*) \cap (\cap_{j=1}^{k} B_{n,k}) \cap C) \ge 1 - \delta/3 - \sum_{j=1}^{k} \delta/2k - \delta/6 = 1 - \delta,$$

equation (23) proves the first statement of the theorem. For the second statement, we have on the same event $A_n(t_*) \cap (\cap_{j=1}^{k} B_{n,k}) \cap C$, and combining the two upper bounds from Lemma 4,

$$\mathcal{E}(\hat{t}_n, \hat{w}_n) \le 4 \cdot n^{-1} \cdot G_n^2(\hat{t}_n) \le 4 \cdot n^{-1} \cdot \sup_{t\in\mathcal{S}_{n,\delta,k}} G_n^2(t) \le 24 \cdot (n\delta)^{-1} \cdot \mathrm{E}[\sup_{t\in\mathcal{S}_{n,\delta,k}} G_n^2(t)],$$

where we used (23) and (24) in the above inequalities, concluding the proof.

# G   Proof of Lemma 2

We prove a slightly more general result, from which Lemma 2 can be immediately deduced.

**Lemma 5.** *Let $n, d \in \mathbb{N}$ and let $\mathcal{T}$ be a finite set. For each $(i,t) \in [n] \times \mathcal{T}$, let $Z_{i,t} \in \mathbb{R}^d$ be random vectors such that for each $t \in \mathcal{T}$, $(Z_{i,t})_{i=1}^n$ are i.i.d. with the same distribution as $Z_t$. For all $t \in \mathcal{T}$, assume that $\mathrm{E}[Z_t] = 0$, and define*

$$\sigma^2(\mathcal{T}) := \sup_{t \in \mathcal{T}} \mathrm{E}\big[\|Z_t\|_2^2\big], \qquad r_n(\mathcal{T}) := \mathrm{E}\left[\sup_{(i,t) \in [n] \times \mathcal{T}} \|Z_{i,t}\|_2^2\right]^{1/2}.$$

*Then*

$$\frac{1}{2} \cdot \frac{\sigma(\mathcal{T})}{n^{1/2}} + \frac{1}{4} \cdot \frac{r_n(\mathcal{T})}{n} \leq \mathrm{E}\left[\sup_{t \in \mathcal{T}} \left\|\frac{1}{n}\sum_{i=1}^n Z_{i,t}\right\|_2^2\right]^{1/2} \leq C(\mathcal{T}) \cdot \frac{\sigma(\mathcal{T})}{n^{1/2}} + C^2(\mathcal{T}) \cdot \frac{r_n(\mathcal{T})}{n},$$

*where $C(\mathcal{T}) := 5\sqrt{1 + \log|\mathcal{T}|}$.*

To prove Lemma 5, we need to recall a few preliminary results. The first is a classical symmetrization inequality, see e.g. [BLM13, Lemma 11.4] or [Wai19, Proposition 4.11] for a proof.

**Lemma 6.** *For each $(i,t) \in [n] \times \mathcal{T}$, let $W_{i,t} \in \mathbb{R}^d$ be random vectors such that for each $t \in \mathcal{T}$, $(W_{i,t})_{i=1}^n$ are i.i.d. with the same distribution as $W_t$. Let $(\varepsilon_i)_{i=1}^n$ be independent Rademacher random variables, independent of the collection of random vectors $W_{i,t}$. Define $\overline{W}_{i,t} := W_{i,t} - \mathrm{E}[W_t]$. Then*

$$\frac{1}{2}\mathrm{E}\left[\sup_{t \in \mathcal{T}}\left\|\frac{1}{n}\sum_{i=1}^n \varepsilon_i \overline{W}_{i,t}\right\|_2^2\right]^{1/2} \leq \mathrm{E}\left[\sup_{t \in \mathcal{T}}\left\|\frac{1}{n}\sum_{i=1}^n \overline{W}_{i,t}\right\|_2^2\right]^{1/2} \leq 2\,\mathrm{E}\left[\sup_{t \in \mathcal{T}}\left\|\frac{1}{n}\sum_{i=1}^n \varepsilon_i W_{i,t}\right\|_2^2\right]^{1/2}.$$

The second result we recall is the Khinchin-Kahane inequality, the specific form we require is obtained from Peña and Giné [PG99, Theorem 1.3.1] by setting $q = p$ and $p = 2$ in that theorem, see also Boucheron et al. [BLM13, page 141].

**Lemma 7.** *For $i \in [n]$, let $z_i \in \mathbb{R}^d$ be fixed vectors. Let $(\varepsilon_i)_{i=1}^n$ be independent Rademacher random variables. Then for all $p \geq 2$,*

$$\mathrm{E}\left[\left\|\sum_{i=1}^n \varepsilon_i z_i\right\|_2^p\right]^{1/p} \leq \sqrt{p-1} \cdot \left(\sum_{i=1}^n \|z_i\|_2^2\right)^{1/2}.$$

A straightforward consequence of Lemma 7 is the following result, which follows from the elementary observation that for a vector $x \in \mathbb{R}^d$, $\|x\|_\infty \leq \|x\|_p \leq d^{1/p}\|x\|_\infty$.

**Lemma 8.** *For $(i,t) \in [n] \times \mathcal{T}$, let $z_{i,t} \in \mathbb{R}^d$ be fixed vectors. Let $(\varepsilon_i)_{i=1}^n$ be independent Rademacher random variables. Then*

$$\mathrm{E}\left[\sup_{t \in \mathcal{T}}\left\|\sum_{i=1}^n \varepsilon_i z_{i,t}\right\|_2^2\right]^{1/2} \leq \frac{5}{2}\sqrt{1 + \log|\mathcal{T}|} \cdot \left(\sup_{t \in \mathcal{T}}\sum_{i=1}^n \|z_{i,t}\|_2^2\right)^{1/2}.$$

*Proof.* Let $p \geq 1$. Then, by Jensen's inequality and Lemma 7

$$\mathrm{E}\left[\sup_{t \in \mathcal{T}}\left\|\sum_{i=1}^n \varepsilon_i z_{i,t}\right\|_2^2\right] \leq \mathrm{E}\left[\left(\sum_{t \in \mathcal{T}}\left\|\sum_{i=1}^n \varepsilon_i z_{i,t}\right\|_2^{2p}\right)^{1/p}\right]$$

$$\leq \left(\sum_{t \in \mathcal{T}}\mathrm{E}\left[\left\|\sum_{i=1}^n \varepsilon_i z_{i,t}\right\|_2^{2p}\right]\right)^{1/p}$$

$$\leq (2p-1) \cdot \left(\sum_{t \in \mathcal{T}}\left\{\sum_{i=1}^n \|z_{i,t}\|_2^2\right\}^p\right)^{1/p}.$$

Recalling that $\|x\|_p \leq d^{1/p}\|x\|_\infty$ for all $x \in \mathbb{R}^d$ and taking $p := 1 + \log|\mathcal{T}|$ yields the result. $\square$

Finally, we need the following consequence of Lemmas 6 and 8. The proof idea is taken from [Tro16].

**Lemma 9.** *For each $(i,t) \in [n] \times \mathcal{T}$, let $W_{i,t} \in \mathbb{R}$ be random variables such that for each $t \in \mathcal{T}$, $(W_{i,t})_{i=1}^n$ are i.i.d. with the same distribution as $W_t$, with $W_t \geq 0$ almost surely. Then*

$$\mathrm{E}\left[\sup_{t \in \mathcal{T}} \sum_{i=1}^n W_{i,t}\right]^{1/2} \leq \left(\sup_{t \in \mathcal{T}} \sum_{i=1}^n \mathrm{E}[W_{i,t}]\right)^{1/2} + 5\sqrt{1 + \log|\mathcal{T}|} \cdot \mathrm{E}\left[\sup_{(i,t) \in [n] \times \mathcal{T}} W_{i,t}\right]^{1/2}.$$

*Proof.* We have by Jensen's inequality and Lemma 6,

$$\mathrm{E}\left[\sup_{t \in \mathcal{T}} \sum_{i=1}^n W_{i,t}\right] \leq \mathrm{E}\left[\sup_{t \in \mathcal{T}} \left|\sum_{i=1}^n W_{i,t} - \mathrm{E}[W_{i,t}]\right|\right] + \sup_{t \in \mathcal{T}} \sum_{i=1}^n \mathrm{E}[W_{i,t}],$$

$$\leq 2\,\mathrm{E}\left[\sup_{t \in \mathcal{T}} \left|\sum_{i=1}^n \varepsilon_i W_{i,t}\right|^2\right]^{1/2} + \sup_{t \in \mathcal{T}} \sum_{i=1}^n \mathrm{E}[W_{i,t}]. \tag{25}$$

Conditioning on the random vectors $W_{i,t}$, we have by Lemma 8 and the assumption $W_{i,t} \geq 0$ a.s.

$$2\,\mathrm{E}\left[\sup_{t \in \mathcal{T}} \left|\sum_{i=1}^n \varepsilon_i W_{i,t}\right|^2\right]^{1/2} \leq 5\sqrt{(1 + \log|\mathcal{T}|)} \cdot \left(\sup_{t \in \mathcal{T}} \sum_{i=1}^n W_{i,t}^2\right)^{1/2},$$

$$\leq 5\sqrt{1 + \log|\mathcal{T}|} \cdot \left(\sup_{(i,t) \in [n] \times \mathcal{T}} W_{i,t}\right)^{1/2} \cdot \left(\sup_{t \in \mathcal{T}} \sum_{i=1}^n W_{i,t}\right)^{1/2}.$$

Taking expectation with respect to $W_{i,t}$, and using the Cauchy-Schwartz inequality yields

$$2\,\mathrm{E}\left[\sup_{t \in \mathcal{T}} \left|\sum_{i=1}^n \varepsilon_i W_{i,t}\right|^2\right]^{1/2} \leq \sqrt{6(1 + \log|\mathcal{T}|)} \cdot \mathrm{E}\left[\sup_{(i,t) \in [n] \times \mathcal{T}} Z_{i,t}\right]^{1/2} \cdot \mathrm{E}\left[\sup_{t \in \mathcal{T}} \sum_{i=1}^n W_{i,t}\right]^{1/2}.$$

Replacing in (25) and solving the resulting quadratic inequality yields the result. $\square$

Equipped with these results, we now prove Lemma 1. The proof idea is taken from [Tro16].

*Proof of Lemma 1.* We start with the lower bound. We have on the one hand

$$\mathrm{E}\left[\sup_{t \in \mathcal{T}} \left\|\frac{1}{n} \sum_{i=1}^n Z_{i,t}\right\|_2^2\right] \geq \sup_{t \in \mathcal{T}} \mathrm{E}\left[\left\|\frac{1}{n} \sum_{i=1}^n Z_{i,t}\right\|_2^2\right] = \sigma^2(\mathcal{T}). \tag{26}$$

On the on other hand, by Lemma 6, we have

$$\mathrm{E}\left[\sup_{t \in \mathcal{T}} \left\|\frac{1}{n} \sum_{i=1}^n Z_{i,t}\right\|_2^2\right]^{1/2} \geq \frac{1}{2} \mathrm{E}\left[\sup_{t \in \mathcal{T}} \left\|\frac{1}{n} \sum_{i=1}^n \varepsilon_i Z_{i,t}\right\|_2^2\right]^{1/2}.$$

Define the random index

$$I \in \operatorname*{argmax}_{i \in [n]} \max_{t \in \mathcal{T}} \|Z_{i,t}\|_2^2.$$

Conditioning on $Z_{i,t}$, we have by Jensen's inequality

$$\mathrm{E}\left[\sup_{t \in \mathcal{T}} \left\|\frac{1}{n} \sum_{i=1}^n \varepsilon_i Z_{i,t}\right\|_2^2\right] \geq \sup_{t \in \mathcal{T}} \mathrm{E}\left[\left\|\mathrm{E}\left[\frac{1}{n} \sum_{i=1}^n \varepsilon_i Z_{i,t}\right]\right\|_2^2\right] = \sup_{t \in \mathcal{T}} \frac{\|Z_{I,t}\|_2^2}{n^2} = \sup_{(i,t) \in [n] \times \mathcal{T}} \frac{\|Z_{i,t}\|_2^2}{n^2},$$

where in the inequality, the outer expectation is with respect to $\varepsilon_I$, and the inner one is with respect to $(\varepsilon_i)_{i \neq I}$. Taking expectation with respect to $Z_{i,t}$ gives

$$\mathrm{E}\left[\sup_{t \in \mathcal{T}}\left\|\frac{1}{n}\sum_{i=1}^{n} Z_{i,t}\right\|_2^2\right]^{1/2} \geq \frac{1}{2} \cdot \frac{r_n(\mathcal{T})}{n} \tag{27}$$

Averaging the lower bounds (26) and (27) yields the desired lower bound. We now turn to the upper bound. We have by Lemmas 6 and 8.

$$\mathrm{E}\left[\sup_{t \in \mathcal{T}}\left\|\frac{1}{n}\sum_{i=1}^{n} Z_{i,t}\right\|_2^2\right]^{1/2} \leq 2\,\mathrm{E}\left[\sup_{t \in \mathcal{T}}\left\|\frac{1}{n}\sum_{i=1}^{n} \varepsilon_i Z_{i,t}\right\|_2^2\right]^{1/2}$$

$$\leq 5\sqrt{1 + \log|\mathcal{T}|} \cdot \mathrm{E}\left[\sup_{t \in \mathcal{T}}\sum_{i=1}^{n}\left\|\frac{1}{n} Z_{i,t}\right\|_2^2\right]^{1/2}$$

Applying Lemma 9 on the last term yields the desired upper bound. □

# H  Proof of Corollary 1

The Glivenko-Cantelli and Donsker assumptions of Theorem 3 follow directly from the moment assumptions of the corollary, the weak law of large numbers, and the central limit theorem, and therefore the conclusions of Theorem 3 hold. For the first statement of the corollary, we may assume without loss of generality that $\mathcal{T}_* \neq \mathcal{T}$, otherwise the statement holds trivially. Define

$$\varepsilon := \min_{t \in \mathcal{T} \setminus \mathcal{T}_*} \{R(t, w_*(t)) - R_*\}.$$

Then $\varepsilon > 0$, and by the first item of Theorem 3,

$$\lim_{n \to \infty} \mathrm{P}\big(\hat{t}_n \notin \mathcal{T}_*\big) \leq \lim_{n \to \infty} \mathrm{P}\big(R(\hat{t}_n, w_*(\hat{t}_n)) - R_* > \varepsilon/2\big) = 0. \tag{28}$$

It remains to prove the improved upper bound on the asymptotic quantiles. For this, referring to the proof of Theorem 3, and in particular to (18), it is enough to show that

$$\lim_{n \to \infty} \mathrm{P}\big(n \cdot \big[R(\hat{t}_n, w_*(\hat{t}_n)) - R_*\big] > 0\big) = 0,$$

but this follows directly from (28).

# I  Proof of Corollary 2

The statement follows from the same argument as Theorem 4 with only a few simple modifications. As explained in the main text, we use Lemma 3 to bound the quantity $\mathrm{E}[\max_{t \in \mathcal{T}} \Lambda_n(t)]$ by constructing a block diagonal matrix. We use Lemma 2 to control, for any subset $\mathcal{S}$, $\mathrm{E}\big[\max_{s \in \mathcal{S}} G_n^2(s)\big]$. The only minor deviation from Theorem 4 is that we bound the second moment

$$\mathrm{E}\left[\sup_{t \in \mathcal{T} \setminus \mathcal{T}_*} \Delta_n^2(t, t_*)\right]$$

instead of the first. This explains the slightly better dependence on $\delta$ in the sample size restriction of Corollary 2 compared to Theorem 4.

