# OpenReview forum: "On the Efficiency of ERM in Feature Learning"
_NeurIPS.cc/2024/Conference — NeurIPS 2024 poster_

### Official Review · Reviewer_bjqG · 2024-07-04

**Soundness:** 4
**Presentation:** 3
**Contribution:** 2
**Rating:** 6
**Confidence:** 4

**Summary:**

This paper considers a "feature learning" setting equivalent to structural risk minimization over a set of hypotheses of the form $\langle w, \phi_t(x)\rangle$. It demonstrate that the statistical error of this procedure converges to a quantity that depends on a natural empirical process defined *only* in terms of the classes which contain an optimal predictor. Thus, it is claimed that ERM is "efficient" at feature learning in a natural way.

EDIT: I raise my score from a 5 to a 6. I encourage the authors to include the counterexample presented to me in the rebuttal in the main text as a form of motivation, and as a didactic example to express "what goes wrong". The authors can also remark about situations under which better dependence on $\delta$ is achievable. Thank you to authors for addressing my points.

**Strengths:**

This paper has a number of strengths that make it compelling.

(1) The formalism studies is **exceptionally** clean and natural. It lends itself to study well in other settings as well.
(2) The authors provide both asymptotic and nonasymptotic results, and the asymptotic ones are rather "sharp" in that they provide insight into both the limiting distributions, and do so in terms of universal, Gaussian empirical processes.
(3) The authors proofs are effective and succinct, and demonstrate great command of the relevant technical machinery.
(4) The authors use of asymptotic bounds allows them to build considerable intuition in the presentation, before introducing the non-asymptotic bounds which are considerably more involving to parse.

**Weaknesses:**

There are a couple weaknesses, however, that temper my excitement.

(1) I apologize for the directness, but I do not find the qualitative finding particularly surprising. We know that ERM localizes, and as a consequence, if I have a predictor class of the form

$\mathcal F= \bigcup_{t \in \mathcal{T}} \mathcal{F}_t$, and all optimal predictors lie in $\mathcal F_t, t \in \mathcal{T}^{\star}$, and moreover, the risk of an  $f \in \mathcal{F}_t$ where, $t \notin \mathcal{T}^{\star}$ is lower bound away from zero, then localization should force the limiting behavior of the problem to only depend on the statistical complexity of

$\bigcup_{t \in \mathcal{T}^\star} \mathcal{F}_t$

It is nice to quantify this rigorously, but again, the phenomenon does not seem to be to bring fundamentally new insight.

(2) The non-asymptotic dependence on the probability of error $\delta$ is quite bad. Indeed, an $O(1/\delta)$ dependence is not even integrable, obviating in expectation bounds (perhaps in-expectation in too much to ask for, given the possibility that the covariance matrices become singular. Still, this seems like a major limitation.

(3) While the problem setting is remarkably clean, Theorem 4 is not. This seems inevitable, and the authors both (a) explain its intuition and (b) instantiate it for a natural class of problems...

(4) I think the authors should include some more intuition about the relevant increments and terms defined in the paper ($G_n(t)$, $\Lambda_n(t)$) and so forth. Citing that "such and such is just such and such in Bos[02]" does not help with intuition building. The authors might consider adding a section in the appendix that elaborates further, and also provides formal definitions of what it means for a class to be Glivenko Cantelli (this was relatively clear) and, more importantly, Donsker (as a reader with the relevant background, I know what is meant, but this could be less accessible to others). The authors might also consider remarking on what limitations the Donkser-ity of the problem entail (e.g. sufficient restriction on metric entropy).

**Questions:**

Can the authors please explain why the localization phenomenon presented in this work is surprising?

**Limitations:**

(1) Weak dependence on error probability $\delta$
(2) Possible limitations due to the Donkser assumption
(3) Restricted to the linear setting

---

> ### Author Rebuttal · Authors · 2024-08-06
>
> We thank the reviewer for engaging with our paper and for their feedback. We address some of their concerns below.
>
> ---
>
> ***I apologize for the directness, but I do not find the qualitative finding particularly surprising. We know that ERM localizes, and as a consequence, if I have a predictor class of the form $\mathcal{F} = \bigcup\_{t \in \mathcal{T}} \mathcal{F}\_{t}$, and all optimal predictors lie in $\mathcal{F}\_{t}$, $t \in \mathcal{T}\_{\*}$, and moreover, the risk of an $f \in \mathcal{F}\_{t}$ where $t \notin \mathcal{T}\_{\*}$ is lower bound away from zero, then localization should force the limiting behavior of the problem to only depend on the statistical complexity of $\bigcup\_{t \in \mathcal{T}\_{\*}}\mathcal{F}\_{t}$. It is nice to quantify this rigorously, but again, the phenomenon does not seem to be to bring fundamentally new insight.***
>
> If one believes on some level that the performance of ERM is related to the complexity of the class on which it is performed,  then our results are quite surprising. Our understanding is that this is a commonly held belief in the machine learning community.
>
> For a reader very familiar with localization, we can see how they would be skeptical that the above statement holds once the setting we propose is presented to them. For such a reader, perhaps the most surprising aspect of our result is that one can get away with extremely weak assumptions on certain empirical processes yet still obtain a strong localization phenomenon. Conversely, and perhaps just as surprising to such a reader, is that without such assumptions, the localization phenomenon can completely vanish.
>
> Here is a counter-example to the reviewer's statement (as we understood it, please correct us if we missed something) that we hope makes this last point clear. Let $\mathcal{F} := \\{x \mapsto \langle w, \phi(x) \rangle \mid w \in \mathbb{R}^{d}\\}$ be a linear class induced by a feature map $\phi$. We can write this $d$-dimensional subspace of functions as the union of its $1$-dimensional subspaces. More explicitly, let $\mathcal{T}$ be the half of the unit Euclidean sphere in $\mathbb{R}^{d}$ with non-negative first coordinate, let $\phi\_{t}(x) := \langle t, x\rangle$ for all $t \in \mathcal{T}$, and set $\mathcal{F}\_{t} := \\{x \mapsto a \cdot \phi\_{t}(x) \mid a \in \mathbb{R}\\}$, then $\mathcal{F} = \bigcup\_{t \in \mathcal{T}} \mathcal{F}\_{t}$. Now consider a well-specified linear regression problem under square loss over $\mathcal{F}$, i.e. $Y = \langle w\_{\*} , \phi(X) \rangle + \varepsilon$ for some $w\_{\*} \in \mathbb{R}^{d}$ and $\varepsilon  \sim \mathcal{N}(0, \sigma^{2})$ independent of $X$. Then the optimal feature map is $t\_{*} = \pm w\_{\*}/ \\|w\_{\*}\\|\_{2}$, and for each $f \in \mathcal{F}\_{t}$ for $t \neq t\_{\*}$, the excess risk of $f$ is lower bounded away from zero, so if the statement of the reviewer held, the excess risk should be of order $1$ (proportional to the complexity of the class corresponding to the optimal feature map $\mathcal{F}\_{t\_{\*}}$, which is one dimensional). Yet, it is a classical result that the excess risk in this problem is of order $d$, i.e. it depends on the complexity of the full class $\mathcal{F}$.
>
> This example shows that the intuitive reasoning that because we have a function class expressed as the union of other function classes we should expect localization might be misleading. More is needed, but perhaps the surprising thing is that not too much is needed: Glivenko-Cantelliness/Donskerity of certain empirical processes.
>
> To a certain extent, we agree that the most significant contribution of our work is to propose a tractable framework where feature learning can be studied rigorously, and we are excited to see how it can be applied to study other problems beyond the supervised learning setting we considered in this paper. Even if our results are not surprising to the reviewer, we hope that the simplicity and usefulness of the framework we introduce and the rigor of our work convinces them that the ideas in this paper are worth sharing.
>
> ***The non-asymptotic dependence on the probability of error is quite bad. Indeed, an $O(1/\delta)$ dependence is not even integrable, obviating in expectation bounds (perhaps in-expectation in too much to ask for, given the possibility that the covariance matrices become singular). Still, this seems like a major limitation.***
>
>
> We note that, as discussed in line 133, under our *very weak assumptions*, this is the best dependence on $\delta$ one can obtain for the performance of ERM. In fact, this bad dependence on $\delta$ has spurred the development of a susbtantial literature that aims at designing procedures with better dependence on $\delta$; please refer to the works cited in line 133.
>
> We chose to present our results under the weakest assumptions possible in order to maximize their range of applicability. We agree with the reviewer that under more stringent assumptions (e.g. boundedness of $X$ and $Y$), our analysis would yield a much improved dependence on $\delta$.

---

> ### Author Response · Authors · 2024-08-06
> **Rebuttal (continued)**
>
> ***I think the authors should include some more intuition about the relevant increments and terms defined in the paper ($G\_{n}(t)$, $\Lambda\_{n}(t)$) and so forth. Citing that "such and such is just such and such in Bos[02]" does not help with intuition building.  The authors might also consider remarking on what limitations the Donkser-ity of the problem entail (e.g. sufficient restriction on metric entropy).***
>
> We will aim at providing more intuition when defining our terms in the final version of our paper. In particular, we will replace the particular passage referred to by the reviewer by the following sentence:
>
> *"The parameter $L$ characterizes the deviation of the supremum of the empirical process $\Lambda\_{n}(t)$ from its mean."*
>
> As for Donskerity, we have refrained from referring to metric entropy in the paper except very briefly in lines (229-230). Our hope was to keep the paper accessible, and as such we preferred to describe Donskerity by analogy to the central limit theorem (lines 157-158). We will briefly mention that Donskerity can be established under appropriate metric entropy restrictions in the main paper, and defer a more in depth discussion to the Appendix.
>
> ***The authors might consider adding a section in the appendix that elaborates further, and also provides formal definitions of what it means for a class to be Glivenko Cantelli (this was relatively clear) and, more importantly, Donsker (as a reader with the relevant background, I know what is meant, but this could be less accessible to others).***
>
> We agree with the reviewer's suggestion, and we will add such a section in the Appendix.

---

> ### Comment · Area_Chair_WhAC · 2024-08-11
> **discussion**
>
> Dear Reviewer bjqG,
>
> Thank you very much for submitting your review report. The author(s) have posted responses to your review. Could you kindly provide comments on whether your concerns have been adequately addressed?
>
> Best regards, AC

---

> > ### Comment · Reviewer_bjqG · 2024-08-13
> > **Following up**
> >
> > Thank you, authors, for your detailed feedback. As indicated in my updated review, I raised my score to reflect that my concerns were addressed.

---

### Official Review · Reviewer_rL5m · 2024-07-06

**Soundness:** 3
**Presentation:** 3
**Contribution:** 1
**Rating:** 4
**Confidence:** 3

**Summary:**

This paper investigates the learning theory of empirical risk minimization (ERM) with feature learning. Under the setting where the optimal finite-sample feature is selected by minimizing the empirical risk over a class of features, the authors show that ERM with feature learning implies convergence of the excess risk under certain assumptions. Moreover, other statistical properties, such as asymptotic normality, are also derived.

**Strengths:**

The statistical theory of this paper regarding feature learning is solid, and the results of the main theorems (Theorem 3 and Theorem 4) seem to be correct, even though I have not had time to check the detailed proofs. In fact, the Glivenko-Cantelli assumptions on the empirical processes and the finite moment assumptions are widely used in learning theory, and the rate $1/n\delta \approx O(1/\sqrt{n})$ seems to be correct.

**Weaknesses:**

1. The title "On the Efficiency of ERM in Feature Learning" seems misleading. Initially, it suggests that ERM should aid feature learning. However, after reading the paper, it appears that the authors are discussing the statistical performance of ERM in the context of feature learning, without clearly demonstrating how ERM aids in feature learning.

2. The authors have reviewed many statistical theory papers. However, in terms of feature learning or representation learning, they didn't conduct a thorough literature review. Specifically, for the final-layer feature, recent studies show that the last-layer feature will converge to an Equiangular Tight Frame (optimal feature for classification problems), which is known as neural collapse, as proposed by Papyan et al. in their paper "Prevalence of neural collapse during the terminal phase of deep learning training." There are some subsequent studies that show that the optimal feature is indeed the minimizer of a regularized ERM. These studies include but are not limited to "Exploring deep neural networks via layer-peeled model: Minority collapse in imbalanced training" (Fang et al., 2021); "A Geometric Analysis of Neural Collapse with Unconstrained Features" (Zhu et al., 2021); and "Neural Collapse in Multi-label Learning with Pick-all-label Loss" (Li et al., 2024). For different loss functions, the authors may refer to "Neural Collapse Under MSE Loss: Proximity to and Dynamics on the Central Path" by Han et al. Regarding learning theory, the sample complexity under neural collapse is investigated by Wang et al. (2024) in their work "Neural Collapse Meets Differential Privacy: Curious Behaviors of NoisyGD with Near-perfect Representation Learning."

3. Some of the claims are confusing and may be over-claimed. Specifically, in the Conclusion, the authors claim that their theory might be used to explain the double descent phenomenon and generalization of deep learning with label noise in Zhang et al., 2021. However, the feature learning setting in this paper is far from being extended to the deep learning setting. Indeed, their assumptions here, such as the moment assumption, should hold for all feature maps in the hypothesis class, which is not verified for deep neural networks. However, existing feature or representation learning theory such as the Neural Collapse theory can partially explain the phenomenon in Zhang et al., such as the overfitted model can still generalize.

4. As the theory is not as enlightening as the authors claimed in their paper since their setting is not practical, this should be regarded as a purely theoretical paper. As a theoretical paper, there should be some room for improvement, such as verifying the assumptions for all feature maps (such as the moment assumptions in Theorem 4) for a certain class. Moreover, as a purely statistical theory paper, a lower bound showing that the obtained finite-sample rate is optimal is necessary for a high-quality publication. It would also be better for the authors to emphasize more technical difficulties compared to ERM without feature learning. In fact, under the Glivenko-Cantelli assumptions and the finite sample assumptions, both the asymptotic theory (Theorem 3) and the non-asymptotic theory (Theorem 4) look like a simple extension of Theorem 1 and Theorem 2 without feature learning, while Theorem 1 and Theorem 2 are not novel in learning theory.

**Questions:**

Please address my concerns in the weaknesses part.

**Limitations:**

Yes

---

> ### Author Rebuttal · Authors · 2024-08-06
>
> We thank the reviewer for taking the time to review our paper. We address some of their concerns below.
>
> ---
>
> ***The authors have reviewed many statistical theory papers. However, in terms of feature learning or representation learning, they didn't conduct a thorough literature review. Specifically, for the final-layer feature, recent studies show that the last-layer feature will converge to an Equiangular Tight Frame (optimal feature for classification problems), which is known as neural collapse,...***
>
> We thank the reviewer for bringing this line of work to our attention; we will carefully discuss their relevance in the final version of the paper. That said, our work is squarely in the realm of statistical learning theory (STL), and as such we have focused our effort on reviewing the STL literature, outlining existing results and techniques, and explaining what is challenging in our newly proposed setting.
>
> ---
>
> ***Some of the claims are confusing and may be over-claimed. Specifically, in the Conclusion, the authors claim that their theory might be used to explain the double descent phenomenon and generalization of deep learning with label noise in Zhang et al., 2021. However, the feature learning setting in this paper is far from being extended to the deep learning setting. Indeed, their assumptions here, such as the moment assumption, should hold for all feature maps in the hypothesis class, which is not verified for deep neural networks. However, existing feature or representation learning theory such as the Neural Collapse theory can partially explain the phenomenon in Zhang et al., such as the overfitted model can still generalize.***
>
> The only two sentences that relate our work to the experiments of [Zha+21] are:
>
> 1- *"The most tantalizing aspect of our results is their **potential** in explaining the experiments in [Zha+21]".*
>
> 2- *"Formally connecting our statements to these experiments is beyond what we achieved here, yet, we believe that the new perspective we took might generate useful insights in this area."*
>
>
> We believe the first sentence above leads to a misinterpretation of what we intended, and we will carefully revise this statement for clarity.
>
> ---
>
> ***As the theory is not as enlightening as the authors claimed in their paper since their setting is not practical, this should be regarded as a purely theoretical paper.***
>
> We agree with the reviewer's second remark that our paper is indeed purely theoretical. However, our work provides a new perspective on the analysis of ERM and feature learning, and has the following main message:
>
>     Learning feature maps is easy when only a small subset of them is good, as the bad ones are quickly discarded by the ERM procedure.
>
> We are not aware of prior works with the same conclusion.
>
> ---
>
> ***It would also be better for the authors to emphasize more technical difficulties compared to ERM without feature learning.***
>
> We have a brief discussion about this in lines 72-78. We will elaborate more on these technical difficulties in the revised paper.
>
> ---
>
>
> ***In fact, under the Glivenko-Cantelli assumptions and the finite sample assumptions, both the asymptotic theory (Theorem 3) and the non-asymptotic theory (Theorem 4) look like a simple extension of Theorem 1 and Theorem 2 without feature learning, while Theorem 1 and Theorem 2 are not novel in learning theory.***
>
> We will clarify the technical contributions and the novelty of the analysis in more detail in the revision. We briefly remark here that
> while Theorem 3 vastly extends Theorem 1, this extension is by no means straightforward:
> 1. It requires a new decomposition (eq. 19) of the excess risk that separates the error coming from the choice of feature map and the error coming from the choice of linear predictor. This is in contrast with Theorem 1 where the feature map is fixed, and there is only one source of error coming from the choice of linear predictor.
> 2. It requires an independent proof that asymptotically, the feature map chosen by ERM is an optimal one. In Theorem 1, we have a single fixed feature map, so there is no analogue to this statement in the proof of Theorem 1.
> 3. It requires a new limiting argument that relies on tools from empirical process theory (limiting gaussian process, continuous mapping theorem) to show that the excess risk is controlled by the complexity of the set of optimal feature maps $\mathcal{T}_{\*}$. The analogous step in the proof of Theorem 1 only requires studying a *single* empirical average, which is readily achieved by an application of the Central limit theorem.
>
> Similarly, while Theorem 4 is an extension of Theorem 2, this extension is again not straightforward:
> 1. It requires a new proof that bounds the suboptimality of the feature map picked by ERM non-asymptotically and relies on a localization argument of Koltchinskii [Kol06]. In Theorem 2, the feature map is fixed, so no analogous step exists in its proof.
> 2. It requires bootstrapping this localization result to show that the excess risk of ERM does not depend on the full complexity of the function class, but rather on the complexity of shrinking subclasses of it. This is done by studying the supremum of a certain empirical process. In Theorem 2, the analogous step requires controlling a *single* empirical average using Markov's inequality.
> 3. It requires showing that the shrinking subclasses of functions whose complexity controls the excess risk of ERM converges to the class of functions induced by the set of optimal feature maps (Lemma 1), recovering the result of Theorem 3 asymptotically. This is also new and unrelated to the proof of Theorem 2 where the feature map is fixed.
>
> ---
>
> We would be happy to answer any questions that may arise during the discussion period.
>
> ---
>
> **References**
>
> [Zha+21]: Zhang, Chiyuan, Samy Bengio, Moritz Hardt, Benjamin Recht, and Oriol Vinyals. "Understanding deep learning (still) requires rethinking generalization."

---

> > ### Comment · Reviewer_rL5m · 2024-08-08
> >
> > 1. The representation learning theory, such as neural collapse, is more related to [Zha+21], as I mentioned. Thus, it is connected to statistical learning theory (STL) since STL aims to derive statistical properties of certain phenomena in machine learning, not just the asymptotic properties or the error bounds. BTW, they can partially explain some phenomena occurring in deep learning, whereas classical (asymptotic or non-asymptotic) learning theory may fail to do so.
> >
> > 2. The most important part I mentioned, the sharpness of the derived bound, is currently not discussed in the rebuttal (maybe due to the limited space?). As I said, once the theoretical framework is formalized, a lower bound that matches the upper bound is essential for a purely theoretical paper, which I believe is common sense in STL (such as in those papers published in AOS).

---

> > > ### Author Response · Authors · 2024-08-09
> > >
> > > We thank the reviewer for their response.
> > >
> > > ---
> > >
> > > ***The representation learning theory, such as neural collapse, is more related to [Zha+21], as I mentioned. Thus, it is connected to statistical learning theory (STL) since STL aims to derive statistical properties of certain phenomena in machine learning, not just the asymptotic properties or the error bounds. BTW, they can partially explain some phenomena occurring in deep learning, whereas classical (asymptotic or non-asymptotic) learning theory may fail to do so.***
> > >
> > > Our work is most closely related to the literature that aims at understanding machine learning procedures through upper and lower bounds on their performance, which we called *statistical learning theory*, but we understand that we might be using the term differently from the reviewer. This will be clarified in the paper once we include a discussion of the additional line of work pointed out by the reviewer.
> > >
> > > Our comments on the relationship between our work and the experiments [Zha+21] are a minor aspect of our paper. They should be seen merely as an invitation to the reader to look at the problem of explaining these experiments through a new perspective.
> > >
> > > ---
> > >
> > > ***The most important part I mentioned, the sharpness of the derived bound, is currently not discussed in the rebuttal (maybe due to the limited space?). As I said, once the theoretical framework is formalized, a lower bound that matches the upper bound is essential for a purely theoretical paper, which I believe is common sense in STL (such as in those papers published in AOS).***
> > >
> > > The second statement of Corollary 1 in our paper provides matching upper and lower bounds on the asymptotic quantiles of the excess risk of ERM, and the gap between these bounds is a factor of *two*.
> > >
> > > The general statement of Theorem 3 only contains an upper bound, but it is straightforward to check that the argument behind the lower bound in Corollary 1 immediately extends to the general case. In this case however, it yields a lower bound on the quantiles of the excess risk of the same form as the upper bound in Theorem 3, but with the supremum replaced by an infimum.
> > >
> > > Roughly speaking, this gap is due to the fact that the sequence of ERMs can “oscillate” between optimal feature maps. This problem already appears if one considers only two features maps which are both optimal, and to the best of our knowledge no matching upper and lower bounds on the quantiles of the excess risk are known even in this simple case.
> > >
> > > On the non-asymptotic front, we note that even in the linear regression case, Theorem 2, there is no known matching lower bound to the upper bound we presented. One may sacrifice interpretability and instead use a tighter upper bound in terms of the quantiles of $\||g(X, Y)\||_{\Sigma^{-1}}^{2}$, which can be reversed up to an absolute constant and a different dependence on $\delta$, under the sample size restriction of the theorem. Moving from the linear regression setting to the case of multiple feature maps we study is more delicate however, and the iterative localization method of [Kol06] we used only yields upper bounds, and sheds little light on lower bounds. Despite this shortcoming, as we have emphasized in the paper, the upper bound we derived in Theorem 4 is asymptotically tight in that it is consistent with the asymptotic behavior of the excess risk we derived in Theorem 3 and Corollary 1.
> > >
> > > If the reviewer thinks the above discussion on lower-bounds is interesting, we would be happy to include it in the final version.

---

> > > > ### Comment · Reviewer_rL5m · 2024-08-09
> > > >
> > > > 1. Please delete the discussion about the potential to explain [Zha+21]. In fact, [Zha+21] shows that even though the training procedure may overfit the training data, the trained model can still generalize (a.k.a., double descent), which motivates the idea of representation learning, such as neural collapse. However, I do not see the potential of using this paper to explain the double descent phenomenon in deep learning.
> > > >
> > > > 2.Regarding optimality, STL focuses more on sample complexity in the non-asymptotic sense, such as the minimax optimal rate, which is also mentioned by the author in the introduction. The minimax lower bound is not provided in the current version, which I believe should be the most interesting aspect of an STL paper.
> > > >
> > > >
> > > > Overall, the derived theory does not significantly contribute to the field of representation learning, and regarding STL, the bound is not minimax optimal. Therefore, I will keep my score as is.

---

### Official Review · Reviewer_pWnU · 2024-07-08

**Soundness:** 3
**Presentation:** 3
**Contribution:** 3
**Rating:** 7
**Confidence:** 3

**Summary:**

This paper consider the problem of regression over the linear classes induced by a collection of feature maps. They study both the asymptotic and the non-asymptotic behavior of the empirical risk minimizer. Surprisingly, although the linear classes has a complexity much higher than that of only one linear map, the authors find that when there is a unique optimal feature map, ERM actually behaves similar with the oracle procedure (knows a priori the optimal feature map). General results for non-unique or even infinite feature map is also provided. The authors also apply their non-asymptotic result on finite feature map cases.

**Strengths:**

The results in this paper is both novel and significant. They show that even though non-optimal feature map exists in training process, the actual upper bound of the excess risk depends on the size of optimal feature maps.
From theoretical perspective, the proofs are solid.
The writing is also good, clearly states the results and the intuition, and how their results improve beyond previous classical results where the feature map set is a singleton. Case study is also provided, giving a comprehensive review of how their general framework can be applied.

**Weaknesses:**

It will be good if more case studies are provided.

**Questions:**

Is it possible to consider some infinite feature map set with some structure, so that you can also compute the constants in your results?

**Limitations:**

No limitations are stated.

---

> ### Author Rebuttal · Authors · 2024-08-06
>
> We thank the reviewer for reading our paper and for their positive comments. We address their question and comments below.
>
> ***It will be good if more case studies are provided.***
>
> As mentioned in another comment, one additional example we wanted to include of some practical relevance is the following. A popular statistical learning problem, which motivates the LASSO procedure, asks the following question: in a regression problem under the square loss, learn the best linear predictor from a $d$-dimensional feature map, knowing that the optimal linear predictor is $k$-sparse, i.e. the optimal weights have only $k$ non-zero entries for $k \ll d$.
>
> Our results allow us to tackle the following more general problem: under no assumptions on the optimal weight vector, learn the best linear predictor from a subset of size $k$ of the $d$ features. This problem fits in our framework neatly by taking $\mathcal{T}$ to be given by the set of all subsets of size $k$ of $[d]$ with $|\mathcal{T}| = {d \choose k}$, and our theorems imply, among other things, that the asymptotic quantiles of the excess risk of ERM on this problem are, up to a factor of two, the same as the oracle procedure that knows a priori the best subset of size $k$ of the $d$ features (assuming it is unique).
>
> ---
>
> ***Is it possible to consider some infinite feature map set with some structure, so that you can also compute the constants in your results?***
>
> As we described in another comment, upper bounds on the various expected suprema appearing in our results can be obtained under an assumption on the metric entropy of the set $\mathcal{T}$ under an appropriately chosen metric. One may then use an $\varepsilon$-net argument along with our finite-case results and an argument that controls the approximation error to obtain an upper bound on these expected suprema. However, obtaining such metric entropy estimates is a highly non-trivial task for any given problem, and requires specialized techniques that leverage the particular structure of the problem. We are currently trying to derive such entropy estimates for two-layer neural networks but we have yet to succeed. We believe that this is a good direction for future work.
>
> ---
> ***Limitations***
>
> Please also note that we discussed the limitations of our work in the last paragraph of the paper. If the reviewer thinks something is missing, we would be happy to discuss it.

---

> > ### Comment · Reviewer_pWnU · 2024-08-11
> >
> > Thanks for your reply.

---

### Official Review · Reviewer_9KPn · 2024-07-10

**Soundness:** 4
**Presentation:** 4
**Contribution:** 3
**Rating:** 7
**Confidence:** 4

**Summary:**

This paper studies the novel setting where we are give  a collection of predefined feature maps (indexed by a set T) and we choose one of these feature maps and then, learn a linear predictor on top of the chosen feature map. The authors derive upper bounds on the excess risk that depend on the number (size) of "optimal" feature maps and not the size of the set T.

**Strengths:**

- The author propose a setting for feature learning that is novel and I find it interesting.

- The result in this setting very satisfying. To the best of my knowledge, this is the first result that shows that the excess risk (or at least the upper bound on the excess risk) depends on the size of "optimal" features.

- The proof outline and strategy seems correct. I have not fully checked all the details of the proofs. But, the ones I have checked are all correct and solid. The paper is also very well written.

**Weaknesses:**

- The analysis ignores the role of the learning algorithm and looks at the problem from a purely statistical perspective. The role of implicit bias of the algorithm is not seen here. On a high level, from the set of optimal features, some features are easier to learn/ achieve than others. This might also shrink the effective size of the features.

- The suprema term in the expression of Theorem 4 is not very interpretable. The case of finite features in the next section makes it more clear. However, I find the finite case not that interesting. Is there an interesting non-finite case that one can analyze to get an interpretable result?

- In general, the features are also learned from data and this gives extra dependencies and goes beyond the setting of this paper (unless some sample splitting is done).

- Can the authors comment on the tightness of Theorem 3 and 4? An the potential challenges of coming up with lower bounds?

- The paper will greatly benefit from a simulation result to support the main finding of the paper. For example, for a simple finite feature case. A theoretical example can also help.

**Questions:**

See weaknesses.

**Limitations:**

The authors adequately addressed the limitations.

---

> ### Author Rebuttal · Authors · 2024-08-06
>
> We thank the reviewer for taking the time to read our paper and for their positive evaluation. We address their comments below.
>
> ---
>
> ***The analysis ignores the role of the learning algorithm and looks at the problem from a purely statistical perspective. The role of implicit bias of the algorithm is not seen here. On a high level, from the set of optimal features, some features are easier to learn/ achieve than others. This might also shrink the effective size of the features.***
>
> At a high-level, we agree. Our guarantees are for the worst-case ERM and it is conceivable that a learning algorithm can avoid such worst-case scenario. Please note however that, as we have briefly hinted at in the conclusion, it is difficult to discuss learning algorithms in our setting given its extreme generality: the index set $\mathcal{T}$ is not even equipped with a topology. One way to study a notion of "(implicit) bias" in our setting is to introduce some ordering on the set $\mathcal{T}$ and consider the procedure that picks the smallest ERM in this ordering. However, we strongly believe that it is more appropriate to study this question for specific model classes and learning algorithms of interest, rather than in the general setting we consider here, as the insights obtained for the above-described procedure might not be transferrable to cases of practical interest.
>
> ---
>
> ***The suprema term in the expression of Theorem 4 is not very interpretable. The case of finite features in the next section makes it more clear. However, I find the finite case not that interesting. Is there an interesting non-finite case that one can analyze to get an interpretable result?***
>
> We agree with the reviewer on this limitation; the current Theorem 4 yields interpretable results only in the finite case. Obtaining interpretable results in the non-finite regime is an interesting and important direction that we leave for future work. One way forward would be to consider for example any set $\mathcal{T}$ satisfying a metric entropy estimate (in an appropriately selected metric). One may then use an $\varepsilon$-net argument along with our finite-case results and an argument that controls the approximation error to obtain an upper bound on the expected suprema of Theorem 4. However this approach only provides an upper bound, and requires some work for it to be practically interesting.
>
> ---
>
> ***Can the authors comment on the tightness of Theorem 3 and 4? An the potential challenges of coming up with lower bounds?***
>
> On the asymptotic side, when the optimal feature map is unique, the second statement of Corollary 1 (of Theorem 3) already provides matching upper and lower bounds on the asymptotic excess risk (and the gap between the bounds is a factor of two).
>
> The general statement of Theorem 3 only contains an upper bound, but it is straightforward to check that the argument behind the lower bound in Corollary 1 immediately extends to the general case. In this case however, it yields a lower bound on the quantiles of the excess risk of the same form as the upper bound in Theorem 3, but with the supremum replaced by an infimum.
>
> Roughly speaking, this gap is due to the fact that the sequence of ERMs can "oscillate" between optimal feature maps. This problem already appears if one considers only two features maps which are both optimal, and to the best of our knowledge no matching upper and lower bounds on the quantiles of the excess risk are known even in this simple case. If the reviewer thinks the above-discussed lower-bound is interesting, we would be happy to include it in the final version.
>
> On the non-asymptotic front, we note that even in the linear regression case, Theorem 2, there is no known matching lower bound to the upper bound we presented. One may sacrifice interpretability and instead use a tighter upper bound in terms of the quantiles of $\||g(X, Y)\||_{\Sigma^{-1}}^{2}$, which can be reversed up to an absolute constant and a different dependence on $\delta$, under the sample size restriction of the theorem. Moving from the linear regression setting to the case of multiple feature maps we study is more delicate however, and the iterative localization method of [Kol06] we used only yields upper bounds, and sheds little light on lower bounds. Despite this shortcoming, as we have emphasized in the paper, the upper bound we derived in Theorem 4 is asymptotically tight in that it is consistent with the asymptotic behavior of the excess risk we derived in Theorem 3 and Corollary 1.
>
> ---
>
> ***The paper will greatly benefit from a simulation result to support the main finding of the paper. For example, for a simple finite feature case. A theoretical example can also help.***
>
> One example we wanted to include of some practical relevance is the following. A popular statistical learning problem, which motivates the LASSO procedure, asks the following question: in a regression problem under the square loss, learn the best linear predictor from a $d$-dimensional feature map, knowing that the optimal linear predictor is $k$-sparse, i.e. the optimal weights have only $k$ non-zero entries for $k \ll d$.
>
> Our results allow us to tackle the following more general problem: under no assumptions on the optimal weight vector, learn the best linear predictor from a subset of size $k$ of the $d$ features. This problem fits in our framework neatly by taking $\mathcal{T}$ to be given by the set of all subsets of size $k$ of $[d]$ with $|\mathcal{T}| = {d \choose k}$, and our theorems imply, among other things, that the asymptotic quantiles of the excess risk of ERM on this problem are, up to a factor of two, the same as the oracle procedure that knows a priori the best subset of size $k$ of the $d$ features (assuming it is unique).
>
> ---
>
> **References**
>
> [Kol06]: Koltchinskii, Vladimir. "Local Rademacher complexities and oracle inequalities in risk minimization."

---

> ### Comment · Reviewer_9KPn · 2024-08-08
>
> I thank the authors for their very thorough response.
>
> The only reason I'm not giving a higher score is that as I discussed in my review, the setting of the paper is very general (as the authors also point out in their paper) and I'm not 100% sure how much the results of this very general setting can shed light on what is happening in practical scenarios. However, this is a very solid paper, giving theoretically neat results for a very general setting, and should be accepted at NeurIPS.

---

### Official Review · Reviewer_qDCv · 2024-07-15

**Soundness:** 3
**Presentation:** 4
**Contribution:** 3
**Rating:** 7
**Confidence:** 4

**Summary:**

The paper considers a linear regression problem where an ERM learner tries to learn a linear predictor and a feature map (over a countable set of feature maps) under some specific assumptions on the set of feature maps and the property of minimizers. The paper analyzes both asymptotic and non-asymptotic behaviors of the excess risk in the problem above and suggests that it matches the rate of the case when we only learn linear predictors on top of a fixed feature map, which is surprising.

**Strengths:**

The paper is well-written and easy to follow. The motivation of this paper is. The key results and directly related results in prior works, along with their intuition behinds, are explained very clearly and concisely. There are a few potential typos (see Weaknesses) but it should not be a problem. The proofs look good to me, though I did not check very carefully.

Overall, I enjoy reading the paper. The only reason why I do not give a higher score is that the setting is too niche (see Weaknesses), which makes the results not so ground-breaking. However, it is still a good paper, and I advocate to accept it.

**Weaknesses:**

The main concern about this paper.
1. The setting is narrow: when I think about feature learning, I imagine we first learn a feature map that gives a good representation of the data. We then want to use the learned feature map that adapts it to a downstream task, which is a linear regression with squared loss in this case. The setting this paper proposed is somewhat the way around: (1) Given a feature map, learn the best linear predictor, (2) Select the best feature map. It seems to me that this paper is considering a non-linear regression problem, not feature learning. More precisely, the paper is trying to solve the problem learn the best feature map specified for the linear regression task. What can we tell about the learned feature map on other downstream tasks? Why the title of the paper is "On the Efficiency of ERM in Feature Learning", instead of "On the Efficiency of ERM in Non-linear Regression"?

2. If we look at the picture from that perspective, prior results (Theorem 1, Theorem 2) seem good enough for me. Given a learned feature map, under some conditions, I have a fast rate of learning a linear predictor. Of course, it is not feature learning at all, but it tells something about the learned feature map on a specific downstream task (linear regression), which is fair enough. There is no need (at least for me) for a result explaining if I learn a linear predictor with squared loss along with a feature map specifically designed for linear regression, what the sample complexity is.

Comments on the Conclusion.
1. The claims on potential explanations for generalization in DNNs: To the best of my knowledge, the reason behind that should be explained by the geometry of the loss landscape of over-parameterized models and the implicit bias of (stochastic) optimization algorithms used. I would not go into detail since it goes beyond the scope of this paper. However, this paper: (1) considers an optimization oracle for ERM, and (2) does not assume any geometry of the set of feature maps indexed by $\mathcal{T}$. Therefore, linking the results in this paper to generalization in DNNs seems unnecessary and inappropriate for me.

Minor comments
1. In the Appendix, it might be helpful if the authors first give a proof sketch for each result for readability.

Minor typos:
 1. Line 164, a comma missing after the inequality.
 2. Line 501, should the RHS be $\frac{1}{2}||\nabla{R(w^*)}||_{\Sigma^{-1}}$?
 3. Multiple commas, dots after (in)equalities missing in the Proof of Theorem 1, 2.

After all, it might be too harsh to undervalue this paper based on the points above. I still think it is a good paper, and the results do not have to be connected to Feature Learning and Generalization to be meaningful.

**Questions:**

See Weaknesses.

**Limitations:**

See Weaknesses.

---

> ### Author Rebuttal · Authors · 2024-08-06
>
> We thank the reviewer for their positive evaluation and detailed feedback. We will make sure to fix the typos for the final version of the paper. We address the reviewer's main concern below.
>
> ---
>
> ***The setting is narrow: when I think about feature learning, I imagine we first learn a feature map that gives a good representation of the data. We then want to use the learned feature map that adapts it to a downstream task, which is a linear regression with squared loss in this case. The setting this paper proposed is somewhat the way around: (1) Given a feature map, learn the best linear predictor, (2) Select the best feature map. It seems to me that this paper is considering a non-linear regression problem, not feature learning. More precisely, the paper is trying to solve the problem learn the best feature map specified for the linear regression task. What can we tell about the learned feature map on other downstream tasks? Why the title of the paper is "On the Efficiency of ERM in Feature Learning", instead of "On the Efficiency of ERM in Non-linear Regression"?***
>
> Please note that our usage of the term "feature learning" in our setting is aligned with recent theoretical work, e.g. [Ba+22, Dam+22, Fre+23].
>
> In plain terms, every predictor in the model classes we consider is jointly determined by a choice of feature map and a linear predictor on top of it, and as such, every learning method that operates on such model classes learns a feature map, i.e. it is performing feature learning.
>
> We agree that other methods exist to learn feature maps outside of the supervised setting we consider, and under different data access models such as transfer learning or self-supervised learning. Nevertheless, this does not exclude our setting, which captures arguably the simplest instantiation of the feature learning idea.
>
> Finally, we agree that it would be nice to obtain performance guarantees within our newly proposed framework in settings other than the supervised learning setup we consider, and we are excited to see what can be done with the new abstractions we introduce in our work. However, given that the framework we propose is new, and studying the performance of ERM on the function classes we introduce already requires the development of new ideas, we leave this to future work.
>
> ---
>
> ***If we look at the picture from that perspective, prior results (Theorem 1, Theorem 2) seem good enough for me. Given a learned feature map, under some conditions, I have a fast rate of learning a linear predictor. Of course, it is not feature learning at all, but it tells something about the learned feature map on a specific downstream task (linear regression), which is fair enough. There is no need (at least for me) for a result explaining if I learn a linear predictor with squared loss along with a feature map specifically designed for linear regression, what the sample complexity is.***
>
> We argue that in the scenario described by the reviewer, one does not know that the **learned** feature map is good. This feature map is itself learned from data through some procedure, which incurs an estimation error. In the simplest case (our case), through ERM on data from the given task, but more generally, through ERM on data from another task in transfer learning as an example, or through contrastive learning in self-supervised learning as another example. Quantifying the estimation error of this learned feature map is an important problem, and part of our work is dedicated to answering this question in our setting (e.g. first statements in Theorems 3 and 4).
>
> ---
>
> ***The claims on potential explanations for generalization in DNNs: To the best of my knowledge, the reason behind that should be explained by the geometry of the loss landscape of over-parameterized models and the implicit bias of (stochastic) optimization algorithms used. I would not go into detail since it goes beyond the scope of this paper. However, this paper: (1) considers an optimization oracle for ERM, and (2) does not assume any geometry of the set of feature maps indexed by $\mathcal{T}$. Therefore, linking the results in this paper to generalization in DNNs seems unnecessary and inappropriate for me.***
>
> We agree with the reviewer that we did not establish a formal link between our results and DNNs; see e.g. lines 336-337
>
> *"Formally connecting our statements to these experiments is beyond what we achieved
> here, yet, we believe that the new perspective we took might generate useful insights in this area."*
>
> We pointed out in a paragraph in the conclusion that our results show a disconnect between the complexity of the function classes we consider (which share the ability to select a feature map in a data-dependent way with DNNs), and the excess risk of ERM on them, which is one of the main surprising empirical findings in the experiments of [Zha+21]. To the best of our knowledge, there is currently no consensus or overwhelming evidence for any of the existing explanations of the results of these experiments, and the paragraph we included is meant as an invitation to the reader to look at the problem from the lense of the new framework we introduced.
>
> If the reviewer believes our wording is misleading, we are open to suggestions on how to make our message clearer.
>
> ---
> **References**
>
> [Ba+22]: Ba, Jimmy, Murat A. Erdogdu, Taiji Suzuki, Zhichao Wang, Denny Wu, and Greg Yang. "High-dimensional asymptotics of feature learning: How one gradient step improves the representation."
>
> [Dam+22]: Damian, Alexandru, Jason Lee, and Mahdi Soltanolkotabi. "Neural networks can learn representations with gradient descent."
>
> [Fre+23]: Frei, Spencer, Niladri S. Chatterji, and Peter L. Bartlett. "Random feature amplification: Feature learning and generalization in neural networks."
>
> [Zha+21]: Zhang, Chiyuan, Samy Bengio, Moritz Hardt, Benjamin Recht, and Oriol Vinyals. "Understanding deep learning (still) requires rethinking generalization."

---

> ### Comment · Reviewer_qDCv · 2024-08-08
> **Reply to the rebuttal**
>
> I thank the authors for the detailed feedback. I am still not convinced with the "feature learning" setting and for me, it is more of a non-linear regression setting. I am also aware of the works that the authors refer to, but to be honest, I do not really like those works and their settings. However, I know that I am being biased, and it also does not affect the contributions of this paper.
>
> As for the conclusion, it would be nice if the authors spent more time discussing the linking between the results and generalization in DNNs in multiple views: (1) how it (potentially) explains generalization (2) the drawback of current settings (and assumptions) and how it conflicts with other potential explanations of generalization in DNNs.
>
> Overall, I still find it a good paper, and I will keep my origin evaluation.

---

> > ### Author Response · Authors · 2024-08-09
> >
> > We thank the reviewer for their feedback and appreciate their comments. We will make use of the additional space in the final version of the paper to address the two points raised by the reviewer.

---

### Decision · Program_Chairs · 2024-09-25

**Decision:**

Accept (poster)

**Comment:**

The paper investigates the performance of empirical risk minimization (ERM) in the context of feature learning. The predicted function first maps the input into the feature space, and then derives the output through a linear mapping. An asymptotic upper bound of the excess risk of ERM, and a non-asymptotic upper bound are derived. The paper is technically sound with rigorous proofs. Concerns, nonetheless, include the narrow setting of the main results (see the review and comments by Reviewers qDCv and rL5m), the lack of a minimax lower bound (see the review and comments by Reviewers rL5m and 9KPn), and the 1/δ term in the non-asymptotic upper bound (see the review and comments by Reviewers 9KPn and bjqG). I strongly encourage the author(s) to take the reviewers' comments inside and address these points in the final version.